# Local-Minima-Preserving Polynomial Relaxation of Ising Problems

**Debraj Banerjee** [1]  **Santanu Mahapatra** [2]  **Kunal N. Chaudhury** [1]

## Abstract

The generalized Ising problem captures a broad spectrum of hard combinatorial problems, including MAX-CUT, Number Partitioning (NPP), and Maximum Independent Set. In this work, we consider the notion of one-flip local minima for this problem. We construct a polynomial relaxation and prove the landscape equivalence theorem: there exists a one-to-one correspondence between the local minima of the relaxation and the one-flip local minima of the original Ising problem. This guarantee reduces the Ising problem to finding the local minima of a smooth function, allowing us to leverage scalable gradient-based optimizers such as ADAM. We demonstrate that our method achieves strong performance across challenging benchmarks, including spin-glass models, MAX-CUT, and NPP.

## 1. Introduction

Combinatorial optimization is fundamental to computer science and operations research, with far-reaching impact across social networks (Yuan et al., 2018), finance (Orús et al., 2019), cryptography (Wang et al., 2020), scheduling (Rieffel et al., 2015), electronic circuit design (Barahona et al., 1988), and biosciences (Kell, 2012; Robert et al., 2021). An important form of combinatorial optimization is the Quadratic Unconstrained Binary Optimization (QUBO) model (Lucas, 2014), which seeks to minimize a quadratic objective $x^\top \mathbf{A} x$ over binary vector $x \in \{0, 1\}^n$. Under the mapping $s \leftarrow 2x - 1$, QUBO is equivalent to the Ising model (Ising, 1925) with the Ising energy $\mathcal{E}(s) = -1/2 s^\top \mathbf{J} s - h^\top s$. This Ising model formulation provides a unified interface for a broad class of NP-hard problems (Karp, 2009), such as the spin-glass model (Sherrington & Kirkpatrick, 1975), Maximum Cut (MAX-CUT),

Maximum Independent Set, and the Number Partitioning Problem (NPP).

**Related Work.** Combinatorial problems exhibit exponential growth in the search space and quickly become impractical for exact methods, such as branch-and-bound solvers (e.g., Gurobi (Gurobi, 2024) and CP-SAT (Perron & Didier, 2025)) at large scales. For the Ising model, most scalable solvers are based on classical heuristics like Simulated Annealing (SA) (Kirkpatrick et al., 1983), which navigate the discrete search space stochastically but often suffer from slow convergence or sampling poor local optima. On the other hand, continuous relaxation methods embed the discrete variables into a continuous domain. Examples include convex (Beck & Teboulle, 2000) and semidefinite relaxation (Luo et al., 2010), notably the Goemans-Williamson algorithm (Goemans & Williamson, 1995) for the MAX-CUT problem (GW-SDP).

In parallel, both physics-inspired and neural-network-based solvers have gained attention for Ising optimization. Physics-inspired approaches include Boltzmann networks (Goto et al., 2018), classical and quantum annealing (Santoro et al., 2002; Yavorsky et al., 2019; Lee et al., 2025; Chen et al., 2025), D-Wave quantum annealer (D-Wave, 2023), free-energy based machine (Shen et al., 2025) as well as Coherent Ising Machines (CIM) (Inagaki et al., 2016; Yamamoto et al., 2017; Tiunov et al., 2019; Honjo et al., 2021) and Simulated Bifurcation Machines (SBM) (Puri et al., 2017; Goto et al., 2019; 2021; Kanao & Goto, 2022; Wang et al., 2023). Message-passing-based iterative solvers have also shown strong asymptotic approximation guarantees on random (spin-glass) Ising models (Montanari, 2025). Machine learning-based approaches have been proposed to handle large-scale Ising instances, including Graph Neural Networks (GNNs) (Schuetz et al., 2022) and convolutional architectures such as the Spring Ising Algorithm (SIA) (Jiang et al., 2024). While promising, these methods often rely on solving complex dynamical systems, carefully tuned annealing schedules, or data-driven priors (Bengio et al., 2021) that may not generalize and typically lack rigorous guarantees on solution quality.

Specifically, continuous relaxations can introduce spurious local minima that appear optimal in the continuous landscape but, after rounding, correspond to suboptimal

---

[1]Department of Electrical Engineering, Indian Institute of Science, Bengaluru, India [2]Department of Electronic Systems Engineering, Indian Institute of Science, Bengaluru, India. Correspondence to: Debraj Banerjee <debrajb@iisc.ac.in>.

*Proceedings of the 43rd International Conference on Machine Learning*, Seoul, South Korea. PMLR 306, 2026. Copyright 2026 by the author(s).

or invalid solutions of the original combinatorial problem. Classical relaxations, such as the GW-SDP for the MAX-CUT problem, have provable approximation guarantees, but are computationally expensive and difficult to scale to large instances, whereas first-order methods (Panageas et al., 2019; Kingma & Ba, 2015) scale efficiently but lack any discrete optimality guarantees (after mapping to the discrete domain).

**Our Work.** In this paper, we leverage scalable first-order optimizers (e.g., ADAM (Kingma & Ba, 2015)) and propose a continuous optimization framework. We introduce the **Mi**nima-**P**reserving **C**ontinuous **R**elaxation of **I**sing **M**odel (**MiP-CRIM**), which preserves discrete local optimality while retaining the scalability of gradient-based optimization. The Python code implementation is available at GitHub[1].

Our approach builds upon important precedents in differentiable optimization. Notably, the recent pCQO-MIS framework (Alkhouri et al., 2025) elegantly demonstrated that for the Maximum Independent Set problem, local minimizers of a specialized quadratic objective correspond exactly to maximal independent sets. We generalize this by establishing a rigorous local-minima equivalence (Theorem 3.13) using a class of attractor functions for general Ising problems.

**Key Contributions.** Our main contributions are:

- **Attractor-based relaxation:** We introduce a physics-inspired family of polynomial regularizers (attractors) obeying some *admissibility conditions*, and relax the Ising problem to a constrained continuous optimization problem.

- **Landscape equivalence theorem:** We use the Hamming distance to characterize discrete (one-flip) local optimality for generalized Ising problems, and under the attractor-based continuous relaxation, prove the *landscape equivalence theorem*: the set of discrete local minima of the Ising problem is equal to the set of local minima of the continuous relaxation (upto scaling by a fixed constant).

- **Gradient-based scalable solver:** We validate our gradient-based solver on challenging large-scale benchmarks for problems including the general Ising model, MAX-CUT, and NPP. Our solver demonstrates superior performance compared to state-of-the-art heuristics, specialized exact solvers, and data-driven methods, effectively recovering high-quality solutions across diverse problem classes.

[1] https://github.com/Phymath0Masics/MiP-CRIM

## 2. Preliminaries

**Notations:** We denote the set of real numbers by $\mathbb{R}$, natural numbers by $\mathbb{N}$, and integers by $\mathbb{Z}$. Vectors are in bold lowercase letters (e.g., $\boldsymbol{x}$, $\boldsymbol{s}$) and matrices in bold uppercase letters (e.g., $\mathbf{J}$, $\mathbf{W}$). $\mathbf{e}_i$ is the vector with all zeros except the $i$-th element equal to 1. For vectors $\boldsymbol{x}, \boldsymbol{y} \in \mathbb{R}^n$, $x_i, y_i$ refer to the $i$-th elements and $\boldsymbol{x} \odot \boldsymbol{y} \in \mathbb{R}^n$ is the Hadamard product with $i$-th element $x_i y_i$. $\mathrm{Diag}(\boldsymbol{x})$ denotes the matrix with diagonal elements from $\boldsymbol{x}$ and non-diagonal elements equal to zero. $\mathbf{I}$ is the identity matrix and $\mathbf{1}$ is the vector of all ones (of appropriate dimension). The sign function $\mathrm{sign}(\cdot)$ operates element-wise on vectors, mapping non-negative values to $+1$ and negative values to $-1$. The projection operator onto a closed set $S$ is denoted by $\mathrm{Proj}_S(\cdot)$, and the indicator function by $\mathbb{1}_{\{\cdot\}}$.

**Ising Model:** The Ising model was introduced by Lenz and Ising as a mathematical model for ferromagnetism (Ising, 1925). In this model, the dimensionless energy function is given by

$$\mathcal{E}(\boldsymbol{s}) = -\frac{1}{2}\boldsymbol{s}^\top \mathbf{J} \boldsymbol{s} - \boldsymbol{h}^\top \boldsymbol{s} \qquad (1)$$

where $\boldsymbol{s} \in \{-1, 1\}^n$ is the spin vector, $\mathbf{J} = \{J_{ij}\}$ is the coupling matrix with $J_{ii} = 0$ (no self-coupling), and $\boldsymbol{h} = \{h_i\}$ is the external field vector.

Solving an Ising model in (1) refers to finding a spin vector (ground state) $\boldsymbol{s}^* \in \{-1, 1\}^n$ that minimizes the Ising energy $\mathcal{E}(\boldsymbol{x}^*)$.

*Remark* 2.1. By introducing an additional spin $s_{n+1}$, we can reformulate (1) as a homogeneous Ising model (with no external field) $-\frac{1}{2}\hat{\boldsymbol{s}}^\top \hat{\mathbf{J}} \hat{\boldsymbol{s}}$. The modified coupling $\{\hat{J}_{ij}\}$ is derived from $\{J_{ij}\}$ and $\{h_i\}$. Since the ground state of (1) can be inferred from this homogeneous model (see Appendix A), it suffices to work with the homogeneous model.

**Reductions:** Many hard combinatorial problems can be reformulated as an Ising problem (Lucas, 2014):

$$\min_{\boldsymbol{s} \in \{-1,1\}^n} \mathcal{E}(\boldsymbol{s}) = -\frac{1}{2}\boldsymbol{s}^\top \mathbf{J} \boldsymbol{s}, \qquad (2)$$

For instance, given an undirected weighted graph $G = (V, E)$ with $|V| = n$ and weighted adjacency matrix $\mathbf{W} \in \mathbb{R}^{n \times n}$, the MAX-CUT problem seeks a bipartition $V = A \cup B$ with $A \cap B = \emptyset$ that maximizes the cut value

$$\mathrm{Cut}(A, B) := \sum_{(i,j) \in E} W_{ij} \mathbb{1}_{\{i \in A, \, j \in B\}}. \qquad (3)$$

Using a spin encoding $\boldsymbol{s} \in \{-1, 1\}^n$ where $s_i = +1$ denotes $i \in A$ and $s_i = -1$ denotes $i \in B$, MAX-CUT can be reformulated as (2) where $\mathbf{J} = -(1/2)\mathbf{W}$ (see Appendix C).

Another example is the NPP, where a multi-set of numbers $N = \{a_1, \ldots, a_n\}$ is given, and the goal is to find a partition $N = A \cup B$ with $A \cap B = \emptyset$ that minimizes the discriminant

$$\text{Disc}(A, B) := \Big| \sum_{a_i \in A} a_i - \sum_{a_i \in B} a_i \Big|. \qquad (4)$$

Let $\mathbf{a} := [a_1, \ldots, a_n]^\top \in \mathbb{R}^n$ and we encode the partition by $\mathbf{s} \in \{-1, 1\}^n$ (e.g., $s_i = +1$ if $a_i \in A$ and $s_i = -1$ if $a_i \in B$). Then minimizing $\text{Disc}(A, B)$ is equivalent to minimizing the quadratic form

$$\Big( \sum_{i=1}^n a_i s_i \Big)^2 = \mathbf{s}^\top (\mathbf{a}\mathbf{a}^\top) \mathbf{s}, \qquad (5)$$

and thus can be posed in the Ising form (2) with

$$\mathbf{J} = -\big(\mathbf{a}\mathbf{a}^\top - \text{Diag}(\mathbf{a} \odot \mathbf{a})\big). \qquad (6)$$

# 3. Minima-Preserving Continuous Relaxation

In this section, we first introduce our continuous relaxation using a class of regularizers to construct the continuous formulation in (9). Next, we provide a rigorous theoretical analysis, establishing the equivalence between local optimality of (9) and discrete (one-flip) local optimality of (2). Finally, we present our differentiable solver MiP-CRIM, which leverages these theoretical guarantees alongside momentum-based gradient methods (such as ADAM).

## 3.1. Continuous Reformulation

The Ising energy $\mathcal{E}(\mathbf{s})$ is homogeneous of degree two, i.e., $\mathcal{E}(\lambda \mathbf{s}) = \lambda^2 \mathcal{E}(\mathbf{s})$ for all $\lambda > 0$. This allows us to extend the discrete search space from $\{-1, 1\}^n$ to the hypercube corners $\mathbb{D} := \big\{\{-\lambda, \lambda\}^n : \lambda > 0\big\}$. We consider the optimization problem

$$\min_{\mathbf{x} \in \mathbb{D}} \mathcal{E}(\mathbf{x}). \qquad (7)$$

A optimizer $\mathbf{x}^*$ of (7) immediately induces a solution of (2) via the mapping $\mathbf{s}^* = \text{sign}(\mathbf{x}^*)$. In other words, problems (2) and (7) are equivalent.

A natural continuous relaxation is obtained by replacing $\mathbb{D}$ with its convex hull, namely the hypercube $[-\lambda, \lambda]^n$, yielding the optimization problem

$$\min_{\mathbf{x} \in [-\lambda, \lambda]^n} \mathcal{E}(\mathbf{x}). \qquad (8)$$

However, solutions of (8) are not guaranteed to lie on $\mathbb{D}$, and hence $\text{sign}(\mathbf{x}^*)$ need not correspond to a meaningful (let alone optimal) solution of the original Ising problem (2).

To bias the continuous relaxation towards the discrete domain $\mathbb{D}$, we introduce a regularizer that acts as an *attractor*

and denote it by $\mathcal{A}$. With the attractor in place, we define the continuous-domain optimization problem

$$\min_{\mathbf{x} \in [-\lambda, \lambda]^n} \mathcal{H}(\mathbf{x}) := \mathcal{E}(\mathbf{x}) + \mathcal{A}(\mathbf{x}), \qquad (9)$$

We refer to the objective $\mathcal{H}$ as the Hamiltonian of (9). We use $\text{loc}(\mathcal{H})$ for the set of local minima of (9).

## 3.2. Universal Class of Attractors

Ideally an attractor must assign equal value to all $\mathbf{x} \in \mathbb{D}$, i.e., $\mathcal{A}(\mathbf{x}) = \mathcal{A}(\mathbf{x}')$ for all $\mathbf{x}, \mathbf{x}' \in \mathbb{D}$. But there are many possible choices of $\mathcal{A}$ satisfying this condition. We focus on a permutation-invariant class that treats all coordinates (spins) symmetrically:

$$\mathcal{A}_f(\mathbf{x}) := \sum_{i=1}^n f_\theta(x_i), \qquad (10)$$

where $f_\theta : \mathbb{R} \to \mathbb{R}$ is a twice differentiable scalar function parametrized by $\theta$ and unbiased at $\pm\lambda$, such that

$$f_\theta(\lambda) = f_\theta(-\lambda), \qquad (\lambda > 0). \qquad (11)$$

Some typical choices for $f_\theta$ are:

$$f_{\theta 1}(x) = -\theta x^2, \quad \theta > 0, \qquad (12)$$

$$f_{\theta 2}(x) = \frac{\beta}{2k+2} x^{2k+2} - \frac{\alpha}{2k} x^{2k}, \qquad (13)$$

$$f_{\theta 3}(x) = \frac{1}{\beta} \int_0^x \tanh^{-1}\Big(\frac{t}{\lambda}\Big) dt - \frac{\alpha}{2} x^2, \qquad (14)$$

where $\theta = (\alpha, \beta, k \in \mathbb{N}) > 0$ for $f_{\theta 2}$, and $\theta = (\alpha, \beta) > 0$, for $f_{\theta 3}$.

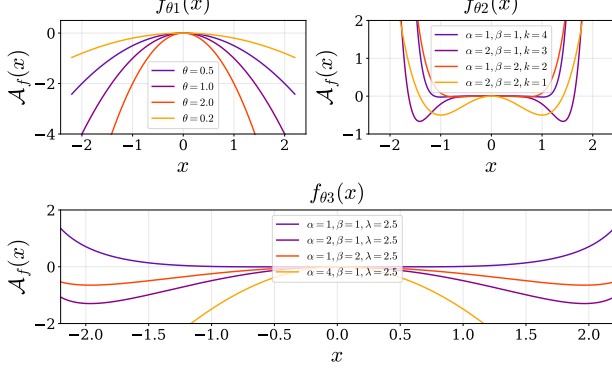

*Figure 1.* Plot of the class of attractor functions: $f_{\theta 1}$, $f_{\theta 2}$ and $f_{\theta 3}$ for various parameter values $\theta$. Both physics-inspired attractors $f_{\theta 2}$ and $f_{\theta 3}$ exhibit a double-well structure, with $\theta$ controlling the depth and curvature of the wells.

Although the quadratic attractor $f_{\theta 1}$ is the simplest choice, the attractors in (13) and (14) are often more suitable: their

double-well geometry (see Figure 1) places the minima *exactly* at the hypercube corners $\mathbb{D}$ (e.g., for (13) the wells occur at $\pm\lambda$ with $\lambda = \sqrt{\alpha/\beta}$). As a result, these regularizers naturally bias the local minima of $\mathcal{E}$ toward the discrete set $\mathbb{D}$. Moreover, all admit direct physical interpretations: $f_{\theta_1}$ works as self-coupling, at $k = 1$, $f_{\theta 2}$ is related to the Hamiltonian of nonlinear parametric oscillators (Goto, 2018), while $f_{\theta 3}$ arises from mean-field approximations commonly used in annealing-based Ising solvers (King et al., 2018) (see Appendix D).

### 3.3. Landscape Equivalence Analysis

We measure the proximity between two spin vectors $s, s' \in \{-1, 1\}^n$ via the Hamming distance (Hamming, 1950):

$$d_H(s, s') := \sum_{i=1}^{n} \mathbb{1}_{\{s_i \neq s'_i\}}, \tag{15}$$

which is the number of spins on which $s$ and $s'$ disagree. Using this metric, we define discrete (one-flip) local optimality (Kammerdiner et al., 2010; Sato et al., 2024) as follows.

**Definition 3.1** (One-flip local minimum). A vector (spin state) $s^* \in \{-1, 1\}^n$ is said to be a local minimum of $\mathcal{E}$ in the 1-nearest-neighborhood sense if

$$\forall s \in \{-1, 1\}^n : d_H(s^*, s) \leqslant 1, \quad \mathcal{E}(s^*) \leqslant \mathcal{E}(s). \tag{16}$$

*Remark* 3.2. For MAX-CUT, a locally optimal spin state $s^*$ induces a partition $(A^*, B^*)$ in (3), such that moving any single vertex from one side of the cut to the other *cannot increase* the cut value. Equivalently, $(A^*, B^*)$ is a *one-vertex locally optimal* cut. For NPP, a locally optimal spin vector $s^*$ yields a partition $(A^*, B^*)$ in (4), such that moving any single number from $A^*$ to $B^*$ (or vice versa) *cannot decrease* the discriminant. Hence, $(A^*, B^*)$ is locally optimal under one-element exchanges.

Although verifying whether $s^*$ is a one-flip local minimum can be done efficiently in $\mathcal{O}(n)$ time, finding such $s^*$ for an arbitrary coupling matrix $\mathbf{J}$ remains computationally challenging. Existing local-search methods are often problem-specific and rely on handcrafted discrete update rules tailored to particular problem formulations.

**Lemma 3.3.** $s^*$ *is a one-flip local minimum of* (2) *if and only if it satisfies the fixed-point condition:*

$$s_i^* = \text{sign}\left((\mathbf{J}s^*)_i\right), \qquad \forall (\mathbf{J}s^*)_i \neq 0. \tag{17}$$

*Proof.* Let $s^* \in \{-1, 1\}^n$ and fix an index $i \in \{1, \ldots, n\}$. Define the one-flip neighbour $s^{(i)} := s^* - 2s_i^* \mathbf{e}_i$. Using (2), the energy difference is $\mathcal{E}(s^{(i)}) - \mathcal{E}(s^*) = 2s_i^*(\mathbf{J}s^*)_i$. $s^*$ is a one-flip local minimum iff $\mathcal{E}(s^{(i)}) - \mathcal{E}(s^*) \geqslant 0$ for all $i$. This implies $s_i^*(\mathbf{J}s^*)_i \geqslant 0$, for all $(\mathbf{J}s^*)_i \neq 0$, which means $s_i^*$ must agree with the sign of $(\mathbf{J}s^*)_i$. $\square$

If for some $i$, $(\mathbf{J}s)_i = 0$, then the value of spin $s_i$ doesn't effect the value of $\mathcal{E}(s)$ and hence in that case we assume $\text{sign}((\mathbf{J}s)_i) = s_i$. Thus we define $\text{loc}(\mathcal{E})$ to be the set of discrete one-flip local minima of (2) as

$$\text{loc}(\mathcal{E}) = \left\{ s \in \{-1, 1\}^n : s = \text{sign}(\mathbf{J}s) \right\}. \tag{18}$$

We next quantify the *margin* by which a corner point violates the discrete stationarity condition (17). For $\mathbf{J} \neq \mathbf{0}$ we define $\gamma(\mathbf{J}) \equiv \gamma$ as

$$\gamma = \min_{s \in \{-1,1\}^n} \left\{ |(\mathbf{J}s)_i| : s_i(\mathbf{J}s)_i < 0 \right\}. \tag{19}$$

By construction, $\gamma > 0$ whenever there exists at least one violating configuration, and $\mathbf{J} \neq \mathbf{0}$ ensures such $\gamma$ exists (see Lemma B.1 in Appendix B).

Computing $\gamma$ exactly via (19) is itself combinatorially hard in general (equivalent to solving $n$ NPP problems). However, for practical implementations using finite precision, we always have the following lower bound.

**Proposition 3.4.** *Let matrix $\mathbf{J}$ be stored with finite precision such that* $\text{lsb}(\mathbf{J}) = 10^{-b}$, *where $b \in \mathbb{Z}$ (lowest significant bit in base 10). Then $\gamma(\mathbf{J}) \geqslant \text{lsb}(\mathbf{J})$.*

*Remark* 3.5. In structured instances like MAX-CUT on unweighted graphs, where the non-zero elements of $\mathbf{J}$ are typically $\pm 1/2$ (or integers), we have the trivial lower bound $\gamma \geqslant 1/2$ (or $\gamma \geqslant 1$).

We now discuss the sufficient conditions on the attractor family $\{\mathcal{A}_f\}$ (equivalently, on $f_\theta$) under which the *landscape equivalence* holds i.e. the local minimizers of the Hamiltonian (9) correspond exactly to the one-flip local minimizers of the discrete Ising model (2). The proofs of all results are available in Appendix B.

**Definition 3.6** (Admissible Attractor Family). For a fixed hypercube length $\lambda > 0$ and a margin $\gamma > 0$ (defined in (19)), a family of twice-differentiable scalar functions $f_\theta$ is called *admissible* if it satisfies:

$$f_\theta(x) = f_\theta(-x) \quad \text{(Symmetry)}, \tag{20}$$

$$f_\theta''(x) < 0, \qquad (|x| < \lambda) \quad \text{(Concavity)}, \tag{21}$$

$$f_\theta'(\lambda) < 0, \quad \text{(Boundary Attraction)} \tag{22}$$

$$f_\theta'(\lambda) + \lambda\gamma > 0. \quad \text{(Robustness)} \tag{23}$$

The symmetry condition in (20) is already satisfied by $f_{\theta 1}, f_{\theta 2}$ and $f_{\theta 3}$. Now, to ensure that the local minimizer $x^*$ of (9) is not a spurious minima and it yields a well-defined discrete solution of (2) via the mapping $\text{sign}(x^*)$, $x^*$ must lie on the hypercube corners $\mathbb{D}$. The following lemma gives a sufficient condition.

**Lemma 3.7.** *Assume* (21) *holds, then every local minimizer of* (9) *lies on the hypercube corners $\mathbb{D} = \{-\lambda, \lambda\}^n$, i.e.*

$$\text{loc}(\mathcal{H}) \subseteq \mathbb{D}. \tag{24}$$

*Remark* 3.8. A direct consequence of Lemma 3.7 is that all stationary points ($x$ where $\nabla \mathcal{H}(x) = 0$) in the interior of $[-\lambda, \lambda]^n$ are non-minimizing (in particular, saddle points), and can be avoided by using random initialization in first-order methods (Panageas et al., 2019). For the quadratic attractor $f_{\theta 1}$ assumption (21) trivially holds, for the quartic attractor $f_{\theta 2}$, it holds if and only if $\alpha > 3\beta\lambda^2$ (for $k = 1$), but for $f_{\theta 3}$ it doesn't hold for any $\theta > 0$.

Now we define the set of points in $\mathbb{D}$ whose signs are discrete one-flip local minima:

$$S_\lambda^* := \{x \in \mathbb{D} : \text{sign}(x) \in \text{loc}(\mathcal{E})\}. \qquad (25)$$

**Lemma 3.9.** *Assume* (20) *and* (22) *hold, then every discrete one-flip local minimizer* $s^* \in \text{loc}(\mathcal{E})$ *induces a local minimizer* $x^* = \lambda s^*$ *of* (9). *Equivalently,*

$$S_\lambda^* \subseteq \text{loc}(\mathcal{H}). \qquad (26)$$

*Remark* 3.10. For the quadratic attractor $f_{\theta 1}$, assumption (22) is trivially satisfied, for the quartic attractor $f_{\theta 2}$, it is equivalent to $\alpha > \beta\lambda^2$ (for $k = 1$), and for $f_{\theta 3}$ it holds when $\alpha\beta\lambda > \tanh^{-1}(1)$.

**Lemma 3.11.** *Assume* (20), (21) *and* (23) *hold, then every local minimizer* $x^* \in \text{loc}(\mathcal{H})$ *induces a discrete one-flip local minimum* $\text{sign}(x^*) \in \text{loc}(\mathcal{E})$, *i.e.*

$$\text{loc}(\mathcal{H}) \subseteq S_\lambda^*. \qquad (27)$$

*Remark* 3.12. For the quadratic attractor $f_{\theta 1}$, the condition (23) becomes $\theta < 0.5\gamma$, and for the quartic attractor $f_{\theta 2}$ with $k = 1$, it becomes $\alpha < \beta\lambda^2 + \gamma$.

Finally, combining Lemma 3.9 and 3.11, we obtain the main equivalence theorem.

**Theorem 3.13.** *Assume* $f_\theta$ *satisfies* (20) *to* (23), *then* $x^* \in \text{loc}(\mathcal{H})$ *if and only if* $\text{sign}(x^*) \in \text{loc}(\mathcal{E})$.

Equivalently, the local minimizers of (9) coincide with the scaled one-flip local minimizers of (2), i.e. $\text{loc}(\mathcal{H}) = S_\lambda^*$.

*Remark* 3.14. The quadratic attractor $f_{\theta 1}$ is admissible iff $0 < \alpha < \gamma$, and for the quartic attractor $f_{\theta 2}$ with $k = 1$, the condition becomes

$$3\beta\lambda^2 < \alpha < \beta\lambda^2 + \gamma. \qquad (28)$$

Building on Lemma 3.3, we define the synchronization score $\text{sync}(s)$ to quantify the proximity of a candidate solution $s \in \{-1, 1\}^n$ to being a one-flip local minimum:

$$\text{sync}(s) := \frac{1}{n} \sum_{i=1}^n \mathbb{1}_{\{s_i = \text{sign}((\mathbf{J}s)_i)\}}. \qquad (29)$$

This metric measures the degree of satisfaction of the fixed-point condition (17), where $\text{sync}(s) = 1$ certifies that $s$ is a stable one-flip local minimum of (2).

## 3.4. Optimization Algorithm

We now translate our theoretical guarantee into a practical, scalable differentiable solver. The core challenge in continuous relaxation is designing a loss landscape where smooth local minima faithfully represent discrete solutions, and selecting an optimization dynamic that efficiently navigates this landscape.

**Choice of Attractor Class.** While our theoretical framework admits any regularizer satisfying conditions (20)–(23), empirical performance varies significantly across classes. As shown in Figure 2a, the quartic attractor $f_{\theta 2}$ consistently yields superior ground state (one-flip local minima) compared to $f_{\theta 1}$ and $f_{\theta 3}$. Hence, we adopt the quartic polynomial attractor for our **MiP-CRIM** (**Mi**nima-**P**reserving **C**ontinuous **R**elaxation for **I**sing **M**odel) algorithm. It leverages standard first-order gradient optimizers to navigate the non-convex landscape. Our experiments (Figure 2b) demonstrate that momentum-based adaptive methods, specifically ADAM (Kingma & Ba, 2015), significantly outperform Momentum Gradient Descent (MGD) (Beck, 2017) and vanilla GD in escaping saddle points and converging to high-quality solutions.

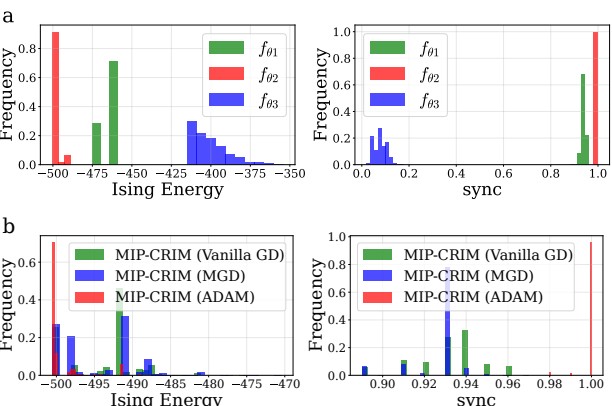

*Figure 2. Comparison of attractor classes on a 100-spin Ising model with 1000 random initializations. (a) The quartic attractor $f_{\theta 2}$ (red) significantly outperforms quadratic $f_{\theta 1}$ (green) and mean-field based $f_{\theta 3}$ (blue) attractors, yielding the lowest Ising energy while achieving synchronization score sync = 1 (29) almost every time. (b) Evaluation of first-order optimizers (used in Algorithm 1) using the quartic attractor $f_{\theta 2}$. ADAM (red) demonstrates superior convergence properties, achieving the best synchronization score (sync = 1) always (along with the lowest Ising energy), each time and consistently lower energy states compared to Momentum Gradient Descent (MGD) and Vanilla GD. The results are obtained using NVIDIA L40S GPU with Intel Xeon 6520P CPU.*

**Implementation Details.** The core operation in $\nabla \mathcal{H}$ is a matrix-vector multiplication ($\mathbf{J}x$), which is highly efficient on modern GPU accelerators. We implement an adaptive grid-search tuning scheme (detailed in Section E) for the

---

**Algorithm 1 MiP-CRIM**

---

**Require:** Coupling matrix $\mathbf{J}$, initialization $\boldsymbol{x}^{\text{in}}$, optimization steps $T$, epochs $K$, hyperparameters $(\alpha, \beta, \lambda)$, Hamiltonian $\mathcal{H}$, Optimizer: Opt, learning rate $\tau$, noise scale $\sigma$

**Ensure:** Best found spin vector $\mathbf{s}^*$

1: $S_{\text{opt}} \leftarrow \emptyset$
2: **for** $k = 1$ to $K$ **do**
3:    $\boldsymbol{x}^{(0)} \leftarrow \boldsymbol{x}^{\text{in}}$
4:    **for** $t = 1$ to $T$ **do**
5:      $\boldsymbol{g}(\boldsymbol{x}^{(t-1)}) \leftarrow \nabla\mathcal{H}(\boldsymbol{x}^{(t-1)})$      (gradient computation)
6:      $\boldsymbol{x}^{(t)} \leftarrow \text{Opt}\left(\boldsymbol{x}^{(t-1)}, \boldsymbol{g}(\boldsymbol{x}^{(t-1)})\right)$    (optimizer)
7:      $\boldsymbol{x}^{(t)} \leftarrow \text{Proj}_{[-\lambda,\lambda]^n}(\boldsymbol{x}^{(t)})$    (box constraint)
8:    **end for**
9:    $\boldsymbol{s}^{(T)} \leftarrow \text{sign}(\boldsymbol{x}^{(T)})$      (spin mapping)
10:   $\boldsymbol{y}^{(T)} \leftarrow \lambda\, \boldsymbol{s}^{(T)} \in \mathbb{D}$     (corner projection)
11:   $\boldsymbol{g}(\boldsymbol{y}^{(T)}) \leftarrow \nabla\mathcal{H}(\boldsymbol{y}^{(T)})$
12:   **if** $\boldsymbol{y}^{(T)} \odot \boldsymbol{g}(\boldsymbol{y}^{(T)}) \leqslant \mathbf{0}$ **then**
13:      $S_{\text{opt}} \leftarrow S_{\text{opt}} \cup \{\boldsymbol{s}^{(T)}\}$   (one-flip local minimum)
14:   **end if**
15:   $\boldsymbol{x}^{\text{in}} \sim \mathcal{N}(\boldsymbol{y}^{(T)}, \sigma\mathbf{I})$    (perturbation for exploration)
16: **end for**
17: **return** $\mathbf{s}^* \in \arg\min_{\boldsymbol{s} \in S_{\text{opt}}} -0.5\, \boldsymbol{s}^\top \mathbf{J}\boldsymbol{s}$

---

hyperparameters $\alpha, \beta, \lambda, \gamma$, and $\tau$ (learning rate). This process ensures that the theoretical admissibility conditions (Eq. 28) are met by the attractor $f_{\theta 2}$ while dynamically refining the search space to locate the optimal parameters for a given problem instance (e.g., MAX-CUT or NPP). Step 12 in Algorithm 1 cheeks if $\boldsymbol{y}^{(T)}$ is a local minima of (9), which ensures $\boldsymbol{s}^{(T)}$ to be the one-flip local minima of (2) by Theorem 3.13. As the saddle points in the interior are strictly unstable (due to $f_\theta'' < 0$ (22)), the optimizer is naturally driven toward the hypercube corners. To escape the saddle points (as discussed in Remark 3.8) and explore the landscape, we employ a "basin-hopping" strategy (Wales & Doye, 1997) using Gaussian perturbations (Step 15 of Algorithm 1).

## 4. Experiments

We evaluate MiP-CRIM algorithm (in Python) on three distinct classes of NP-hard combinatorial optimization problems: the Sherrington-Kirkpatrick (SK) spin glass model (Sherrington & Kirkpatrick, 1975), MAX-CUT, and NPP using the reduction formulations described in Section 2. All experiments were conducted on a workstation equipped with a 44GB NVIDIA L40S GPU and an Intel Xeon 6520P CPU. We compare our solver against a comprehensive suite of baselines, including classical exact solvers (Gurobi (Gurobi, 2024), CP-SAT (Perron & Didier, 2025)),

the GW-SDP relaxation (GW-SDP (Goemans & Williamson, 1995)), quantum annealing (D-WAVE (D-Wave, 2023)), and state-of-the-art physics-inspired heuristics: the Ballistic Simulated Bifurcation Machine (bSBM (Goto et al., 2021)), data-dependent and GNN based PI-GNN (Schuetz et al., 2022), Point-CNN based Spring Ising Algorithm (SIA (Jiang et al., 2024)), Neuromorphic Simulated Annealing (NeuroSA (Chen et al., 2025)), and the Free Energy Machine (FEM (Shen et al., 2025)).

**Spin-Glass Model.** We first benchmark optimization performance on the Sherrington-Kirkpatrick (SK) model with $n = 10^3$ spins, a canonical dense spin-glass instance with couplings $J_{ij} = J_{ji}$ drawn from a standard normal distribution. We use 100 independent runs for each solver to get the best and mean Ising energy values. As shown in Table 1 (left panel), MiP-CRIM achieves the second-best result behind FEM while significantly outperforming classical annealing and GW-SDP. Notably, our solver maintains a perfect synchronization score (sync = 1.0), matching D-WAVE and FEM, which indicates that the solver successfully converges to a consistent discrete configurations (one-flip local minima) without oscillation. In Table 2 we benchmark with Incremental Approximate Message Passing (IMAP) (Montanari, 2025) which is SOTA for the Gaussian Orthogonal Ensemble (GOE) model, and comes with $(1 - \epsilon)$-approximate solution with high probability. MiP-CRIM consistently achieves lower Ising energy than IMAP on both GOE and standard SK models, while converging to valid one-flip local minima (sync = 1) across all runs.

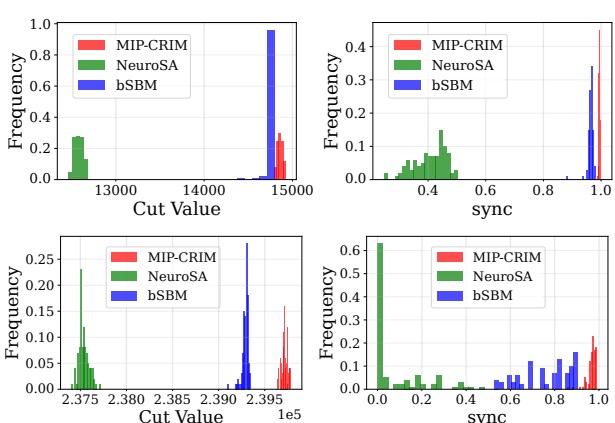

*Figure 3.* MAX-CUT comparison on Erdős–Rényi (ER) graphs. Top row: sparse ER graph with $n = 1000$ and edge probability $p = 0.05$; bottom row: dense ER graph with $n = 1000$ and $p = 0.95$. The left column shows the distribution of achieved cut values, while the right column shows the distribution of the synchronization score: sync (29). Each solver (MiP-CRIM with ADAM optimizer, NeuroSA, and bSBM) is run 100 times with independent random initializations to generate the histograms. We have used NVIDIA L40S GPU with Intel Xeon 6520P CPU system.

| Solver | 1000 spin SK Model | | | K2000 Graph | | | NPP ($n = 1000$) | | |
|---|---|---|---|---|---|---|---|---|---|
| | Energy ($\downarrow$) | sync ($\uparrow$) | Runtime (s) | Cut Val ($\uparrow$) | sync ($\uparrow$) | Runtime (s) | Disc ($\downarrow$) | sync ($\uparrow$) | Runtime (s) |
| GW-SDP | -15053.04 | 0.937 | 155.54 | 32018 | 0.945 | 1144.58 | 1.414 | 0.489 | 88.97 |
| | -14453.04 | 0.907 | | 31697 | 0.917 | | 10.468 | 0.310 | |
| bSBM (Goto et al., 2021) | -16634.97 | 0.997 | 0.12 | 33473 | 0.998 | 0.17 | 0.0123 | 0.991 | 0.12 |
| | -16357.12 | 0.993 | | 32993.66 | 0.996 | | 0.8785 | 0.675 | |
| D-WAVE (D-Wave, 2023) | -16593.14 | **1.000** | 0.98 | 33775 | **1.000** | 9.64 | 0.00957 | 0.997 | 1.073 |
| | -16471.64 | 0.995 | | 32194.87 | 0.996 | | 1.191 | 0.973 | |
| SIA (Jiang et al., 2024) | 16350.10 | **1.000** | 0.13 | 32467 | **1.000** | 0.19 | 0.25 | 0.903 | 0.14 |
| | -16037.15 | 0.995 | | 32433.87 | 0.996 | | 16.67 | 0.502 | |
| NeuroSA (Chen et al., 2025) | -14106.47 | 0.912 | 27.07 | 32741 | 0.977 | 39.36 | 0.176 | 0.488 | 28.57 |
| | -13989.41 | 0.879 | | 31598.87 | 0.959 | | 0.579 | 0.371 | |
| FEM (Shen et al., 2025) | **-16814.75** | **1.000** | **0.01** | 34038 | **1.000** | **0.01** | 313.87 | 0.132 | **0.01** |
| | **-16593.16** | 0.994 | | 3306.66 | 0.997 | | 514.86 | 0.017 | |
| **MiP-CRIM (ours)** | -16689.49 | **1.000** | 0.21 | **34137** | **1.000** | 0.24 | **0.0015** | **1.000** | 0.21 |
| | -16491.62 | **0.999** | | **33473.33** | **0.998** | | **0.014** | **0.994** | |

*Table 1. Comparative performance on large-scale NP-hard benchmarks: 1000-spin SK Model, K2000 Graph (MAX-CUT), and 1000 uniformly generated numbers in $[0, 1]$ (NPP). We report the best (top) and mean (bottom) values over 100 runs for the primary objectives: Ising Energy (2), Cut Value (3), and Partition Discriminant (4), alongside the ground state synchronization score sync (29) against runtime (seconds). Baselines include the classical GW-SDP (Goemans & Williamson, 1995), physics-inspired solvers (bSBM (Goto et al., 2021), SIA (Jiang et al., 2024)), quantum annealing (D-WAVE (D-Wave, 2023)), neuromorphic annealing NeuroSA (Chen et al., 2025) and the latest free energy based FEM (Shen et al., 2025)). **Bold** values indicates the best result, and the underlined values indicates the second best. ADAM optimizer is used inside MiP-CRIM solver. All solvers are tuned to give the best results (see Appendix E), and the runtimes exclude the hyper-parameter tuning. The results are obtained on an NVIDIA L40S GPU and an Intel Xeon 6520P system.*

**MAX-CUT.** We assess MAX-CUT performance across three graph topologies: dense complete graphs $K_n$ (with $n$ nodes and edge weights independently sampled from $\pm 1$ with equal probability), Erdős–Rényi graphs (ERDdS & R&wi, 1959) $\text{ER}(n, p)$ (with $n$ nodes and each edge is included independently with probability $p$.), and the standard G-set[2] benchmark. On the $K_{2000}$ instance (Table 1, middle panel), MiP-CRIM identifies the best cut value among all solvers, surpassing both D-WAVE and FEM within a highly competitive runtime.

For the standard G-set benchmarks (Table 3), MiP-CRIM consistently finds solutions within $1\%$ relative error of the best-known results. It also beats the graph-neural-net based PI-GNN and matching the optimal known cut for the large 3000-node graph G49 and outperforms D-WAVE on the instances: G14, G50, and G55. Results on other G-set graphs and the parameter tuning process (along with optimal parameter values) are detailed in Appendix F.1.

**Number Partitioning Problem (NPP).** Finally, we evaluate the solver on the NPP with $n = 1000$ numbers drawn from uniform $[0, 1]$ distribution. This problem is characterized by extreme sensitivity to precision. Table 1 (right panel) reports the partition discriminant (4). MiP-CRIM achieves the lowest discriminant among all heuristics, significantly outperforming the best alternative D-WAVE and FEM . This

[2]https://web.stanford.edu/~yyye/yyye/Gset/

result underscores the efficacy of our continuous relaxation scheme in handling problems with stiff, high-precision constraints.

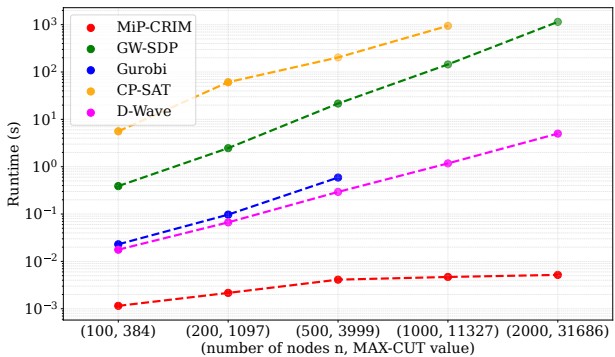

*Figure 4. Runtime comparison for MAX-CUT on complete graphs. Complete graphs with $n$ nodes and randomly sampled $\pm 1$ edge weights are considered. The x-axis reports $(n, \text{MAX-CUT})$, where the MAX-CUT value is obtained by 1000 random roundings of the GW-SDP. The y-axis shows the runtime required by each solver to match this reference value. ADAM is used in the MiP-CRIM algorithm. For $n > 500$, Gurobi and CP-SAT fail to scale and are therefore omitted. We have used NVIDIA L40S GPU with Intel Xeon 6520P CPU system.*

## 5. Discussion

**Scalability and Robustness.** The computational scalability of MiP-CRIM is highlighted in Figure 4. While ex-

| Type | Spins | Avg Ising Energy | | Best Ising Energy | | Avg sync / Time (s) | |
|---|---|---|---|---|---|---|---|
| | | IAMP | MiP-CRIM | IAMP | MiP-CRIM | IAMP | MiP-CRIM |
| GOE | 100 | -62.00 | **-71.39** | -70.72 | **-77.46** | 0.938 / 0.037 | **1.000** / 0.022 |
| GOE | 200 | -127.76 | **-144.74** | -138.11 | **-155.24** | 0.941 / 0.043 | **1.000** / 0.034 |
| GOE | 500 | -322.67 | **-363.59** | -338.25 | **-375.80** | 0.944 / **0.071** | **1.000** / 0.091 |
| GOE | 1000 | -643.85 | **-728.27** | -664.81 | **-750.32** | 0.943 / 0.179 | **1.000** / 0.137 |
| SK | 100 | -414.84 | **-504.95** | -515.43 | **-545.49** | 0.915 / 0.035 | **1.000** / 0.022 |
| SK | 200 | -1122.42 | **-1438.63** | -1347.49 | **-1542.79** | 0.900 / 0.039 | **1.000** / 0.035 |
| SK | 500 | -4129.83 | **-5752.52** | -4563.74 | **-5998.39** | 0.880 / **0.060** | **1.000** / 0.088 |
| SK | 1000 | -10895.32 | **-16269.86** | -11660.55 | **-16767.25** | 0.870 / 0.144 | **1.000** / 0.138 |

*Table 2.* Comparison of IAMP and MiP-CRIM on GOE and SK models. GOE: $J_{ij} = J_{ji} \sim \mathcal{N}(0, 1/n)$, $J_{ii} \sim \mathcal{N}(0, 2/n)$; SK: $J_{ij} = J_{ji} \sim \mathcal{N}(0, 1)$, $J_{ii} = 0$. $J_{ij}$ are quantized to 5 decimals ($\gamma_0 = 10^{-5}$). MiP-CRIM uses fixed parameters tuned only on a 100-spin instance satisfying admissibility ($3\beta\lambda^2 < \alpha < \beta\lambda^2 + \gamma_0$). IAMP uses best-tuned parameters per model. Best and mean values are obtained over 100 instances. Results show MiP-CRIM consistently achieves lower energies and perfect synchronization score (sync = 1) with comparable or better runtime.

| G-set | Nodes | Best-Known | D-WAVE | FEM | PI-GNN | MiP-CRIM | Rel. Err. (%) |
|---|---|---|---|---|---|---|---|
| G14 | 800 | **3064** | 3056 | 3024 | 3026 | 3056 | 0.26 |
| G15 | 800 | **3050** | 3046 | 3003 | 2990 | 3037 | 0.43 |
| G22 | 2000 | **13359** | 13357 | 13155 | 13181 | 13326 | 0.25 |
| G49 | 3000 | **6000** | **6000** | 5564 | 5918 | **6000** | 0.00 |
| G50 | 3000 | **5880** | 5852 | 5590 | 5820 | 5856 | 0.41 |
| G55 | 5000 | **10299** | 10259 | 9855 | 10138 | 10173 | 1.22 |
| G70 | 10000 | **9591** | 9514 | 8938 | 9421 | 9531 | 0.21 |

*Table 3. Comparison of MaxCut values on G-set graphs.* The best-known cut values are taken from (Ma & Hao, 2017) and PI-GNN results are from (Schuetz et al., 2022). Best results are highlighted in **bold**, while second-best results are underlined. FEM, and MiP-CRIM (with ADAM optimizer) are tuned for optimal performance. The relative error values at the right-most column are calculated for the cuts ($c_1$) obtained by MiP-CRIM w.r.t. the best-known cut values ($c_2$) as $100 \times (c_2 - c_1)/c_2$. The experiments are conducted on an NVIDIA L40S GPU and an Intel Xeon 6520P system.

act solvers like Gurobi and CP-SAT fail to scale beyond $n = 500$ nodes (exceeding reasonable time limits), MiP-CRIM exhibits flat, near-constant runtime scaling, and solving large-scale instances orders of magnitude faster than GW-SDP and D-WAVE. Furthermore, Figure 3 demonstrates the solver's robustness on $n = 1000$ node ER graphs. On both sparse ($p = 0.05$) and dense ($p = 0.95$) topologies, where MiP-CRIM produces a tightly concentrated distribution of high cut values. The histograms of the synchronization score sync (Figure 3, right column) confirm that MiP-CRIM consistently achieves near-perfect one-flip local minima (sync $\approx 1.0$), contrasting sharply with the higher variance observed in NeuroSA.

MiP-CRIM requires only lightweight parameter tuning via an adaptive grid search with 3 to 5 values per parameter (Appendix E). Importantly, tuning is performed once per problem class (e.g., SK, GOE, K2000), and the resulting configuration remains robust across different instances within that class, as evidenced by Figure 3 and Table 1. In practice, parameters can be calibrated on small-scale instances and reliably transferred to larger problems, enabling scalability and real-time tuning with minimal overhead (as shown in Table 2).

The experimental results indicate that the proposed relaxation produces reliable solutions and captures key aspects of the underlying combinatorial structure. Across a range of problem settings, the method consistently yields stable and meaningful solutions, suggesting that the relaxation aligns well with the original discrete problems in practice.

**Stability and Consistency.** Across all benchmarks, we consistently observe near-perfect synchronization scores (sync $\approx 1.0$ in Table 1, 2 and Figure 3), indicating strong alignment between continuous solutions and discrete spin configurations. In contrast to many heuristic relaxations, where continuous minimizers often drift away from discrete solutions and require additional rounding or post-processing, the polynomial attractor $f_{\theta_2}$ used in MiP-CRIM naturally shapes the optimization landscape so that stable points lie directly on discrete configurations. This behaviour aligns with the theoretical guarantee in Theorem 3.13.

The results on the Number Partitioning Problem (NPP) are particularly informative. NPP is highly sensitive to numerical precision, as small continuous errors can lead to large discriminants in the final partition. MiP-CRIM performs well even under these stringent conditions and, in several

cases, surpasses specialized solvers. This suggests that the polynomial attractors provide a stable mechanism for handling fine-grained constraints, which can be challenging for data-driven methods like PI-GNN. At the same time, the runtime behaviour on MAX-CUT (Figure 4) indicates that this improved precision does not compromise scalability: the solver retains the favourable scaling properties typical of first-order gradient-based methods.

**Future Directions.** There are many natural directions for future work. The gradient-based nature of MiP-CRIM makes it a good fit for emerging analog hardware, such as optical Coherent Ising Machines (Wang et al., 2013). Extending the framework to handle hard constraints directly could broaden its use in applications like scheduling and routing. Another promising direction is to learn or adapt the class of admissible attractors automatically while retaining the theoretical guarantees.

## 6. Conclusion

MiP-CRIM is a continuous optimization framework for general Ising problems that preserves one-flip local optimality under relaxation. In practice, MiP-CRIM proves to be both versatile and robust, while highly specialized solvers may offer small advantages on particular instances, our method performs reliably across a wide range of problems, from sparse G-set graphs to the numerically challenging Number Partitioning task where many baselines struggle. MiP-CRIM also scales efficiently on modern GPUs, enabling the solution of large instances that are beyond the reach of exact solvers such as Gurobi or CP-SAT. By combining scalable gradient-based optimization with local-optimality guarantees for Ising formulations, MiP-CRIM provides a unified and practical approach to solving diverse NP-hard problems.

## Impact Statement

We introduce MiP-CRIM, a continuous relaxation for the Ising problems, that comes with a theoretical guarantee. This has significant potential to accelerate large-scale problem solving in high-impact domains such as logistics, circuit design, and operations research, where NP-hard problems like MAX-CUT and Number Partitioning are ubiquitous. MiP-CRIM offers an energy-efficient alternative to specialized platforms like hardware-based Ising machines and quantum annealers.

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

# Appendix

In this appendix, we first present the reduction to the homogeneous Ising model (Appendix A) and provide detailed proofs of our theoretical results (Appendix B). We then review the MAX-CUT formulation and the Goemans-Williamson SDP relaxation (Appendix C), and discuss how our attractor functions relate to nonlinear parametric oscillators and mean-field annealing (Appendix D). Appendix E describes our adaptive hyperparameter tuning strategy, and we conclude with additional G-set experiments and ablation studies in Appendix F.

## A. Reduction to the Homogeneous Ising Model

In this section, we demonstrate the standard technique used to transform a general $n$-spin Ising model with linear terms (external field) into an equivalent $(n + 1)$-spin homogeneous Ising model (quadratic terms only). This reduction allows us to treat all problem instances using the unified homogeneous formulation $\min -\frac{1}{2}\boldsymbol{s}^\top \mathbf{J}\boldsymbol{s}$ used throughout the main text.

Consider the general Ising energy function with an external field vector $\boldsymbol{h} \in \mathbb{R}^n$:

$$\mathcal{E}(\boldsymbol{s}) = -\frac{1}{2}\boldsymbol{s}^\top \mathbf{J}\boldsymbol{s} - \boldsymbol{h}^\top \boldsymbol{s}, \qquad \boldsymbol{s} \in \{-1, 1\}^n. \tag{30}$$

To absorb the linear term $-\boldsymbol{h}^\top \boldsymbol{s}$, we introduce an auxiliary spin $s_{n+1}$ (denoted here as $t$) and define the augmented state vector $\boldsymbol{\sigma} = [\boldsymbol{s}^\top, t]^\top \in \{-1, 1\}^{n+1}$. We construct the augmented coupling matrix $\hat{\mathbf{J}} \in \mathbb{R}^{(n+1)\times(n+1)}$ as:

$$\hat{\mathbf{J}} = \begin{pmatrix} \mathbf{J} & \boldsymbol{h} \\ \boldsymbol{h}^\top & 0 \end{pmatrix}. \tag{31}$$

The corresponding homogeneous energy function is given by:

$$\hat{\mathcal{E}}(\boldsymbol{\sigma}) = -\frac{1}{2}\boldsymbol{\sigma}^\top \hat{\mathbf{J}}\boldsymbol{\sigma}, \qquad \boldsymbol{\sigma} \in \{-1, 1\}^{n+1}. \tag{32}$$

**Proposition A.1.** *Let $\boldsymbol{\sigma}^* = (\boldsymbol{s}_0, t_0)$ be a ground state of the homogeneous model (32), where $\boldsymbol{s}_0 \in \{-1, 1\}^n$ is the sub-vector of the first $n$ spins and $t_0 \in \{-1, 1\}$ is the auxiliary spin. Then, the vector $\boldsymbol{s}^* = t_0 \boldsymbol{s}_0$ is a ground state of the original model with external field (30).*

*Proof.* First, we expand the homogeneous energy $\hat{\mathcal{E}}$ for an arbitrary augmented state $(\boldsymbol{s}, t)$:

$$\begin{aligned}
\hat{\mathcal{E}}((\boldsymbol{s}, t)) &= -\frac{1}{2}\begin{pmatrix} \boldsymbol{s}^\top & t \end{pmatrix}\begin{pmatrix} \mathbf{J} & \boldsymbol{h} \\ \boldsymbol{h}^\top & 0 \end{pmatrix}\begin{pmatrix} \boldsymbol{s} \\ t \end{pmatrix} \\
&= -\frac{1}{2}\left(\boldsymbol{s}^\top \mathbf{J}\boldsymbol{s} + t\boldsymbol{s}^\top \boldsymbol{h} + t\boldsymbol{h}^\top \boldsymbol{s}\right) \\
&= -\frac{1}{2}\boldsymbol{s}^\top \mathbf{J}\boldsymbol{s} - t\boldsymbol{h}^\top \boldsymbol{s}.
\end{aligned} \tag{33}$$

Next, consider the energy of the candidate solution $\tilde{\boldsymbol{s}} = t\boldsymbol{s}$ in the original model (30):

$$\begin{aligned}
\mathcal{E}(t\boldsymbol{s}) &= -\frac{1}{2}(t\boldsymbol{s})^\top \mathbf{J}(t\boldsymbol{s}) - \boldsymbol{h}^\top(t\boldsymbol{s}) \\
&= -\frac{1}{2}t^2 \boldsymbol{s}^\top \mathbf{J}\boldsymbol{s} - t\boldsymbol{h}^\top \boldsymbol{s}.
\end{aligned} \tag{34}$$

Since $t \in \{-1, 1\}$, we have $t^2 = 1$. Thus, we establish the exact equivalence:

$$\mathcal{E}(t\boldsymbol{s}) = \hat{\mathcal{E}}((\boldsymbol{s}, t)). \tag{35}$$

By definition, if $\boldsymbol{\sigma}^* = (\boldsymbol{s}_0, t_0)$ is the ground state of $\hat{\mathcal{E}}$, then $\hat{\mathcal{E}}((\boldsymbol{s}_0, t_0)) \leqslant \hat{\mathcal{E}}((\boldsymbol{s}, 1))$ for any $\boldsymbol{s} \in \{-1, 1\}^n$. Using the equivalence relation (35):

$$\mathcal{E}(t_0 \boldsymbol{s}_0) = \hat{\mathcal{E}}((\boldsymbol{s}_0, t_0)) \leqslant \hat{\mathcal{E}}((\boldsymbol{s}, 1)) = \mathcal{E}(1 \cdot \boldsymbol{s}) = \mathcal{E}(\boldsymbol{s}). \tag{36}$$

This inequality holds for all $\boldsymbol{s} \in \{-1, 1\}^n$, implying that $\boldsymbol{s}^* = t_0 \boldsymbol{s}_0$ minimizes the original energy $\mathcal{E}(\boldsymbol{s})$. $\qquad \square$

## B. Mathematical Proofs

**Lemma B.1.** *Let $\mathbf{J} \in \mathbb{R}^{n \times n}$ be a coupling matrix ($\mathbf{J}^\top = \mathbf{J}$, $J_{ii} = 0$ for all $i$). Suppose that*

$$s \odot (\mathbf{J}s) \;\geqslant\; \mathbf{0} \qquad \text{for every } s \in \{-1,1\}^n. \tag{37}$$

*Then $\mathbf{J} = \mathbf{0}$.*

*Proof.* Fix $i \in \{1, \dots, n\}$. The $i$-th component of $s \odot (\mathbf{J}s)$ is

$$[s \odot (Js)]_i \;=\; s_i \sum_{j=1}^{n} J_{ij} s_j \;=\; \sum_{j \neq i} J_{ij}\, s_i s_j,$$

where the last equality uses $J_{ii} = 0$. Replacing $s$ by $-s$ leaves each product $s_i s_j$ unchanged, so the constraint (37) at $s$ and at $-s$ coincide. We can also restrict to those $s$ with $s_i = 1$ without loss of generality, and (37) applied to the $i$-th coordinate becomes

$$\sum_{j \neq i} J_{ij}\, \varepsilon_j \;\geqslant\; 0 \qquad \text{for every } \varepsilon \in \{-1,1\}^{\{1,\dots,n\} \setminus \{i\}}. \tag{38}$$

Let $v \in \mathbb{R}^{n-1}$ denote the vector with entries $J_{ij}$ for $j \neq i$, so that (38) reads $\langle v, \varepsilon \rangle \geqslant 0$ for all sign vectors $\varepsilon$. Applying the same inequality to $-\varepsilon$ yields $\langle v, \varepsilon \rangle \leqslant 0$, hence

$$\langle v, \varepsilon \rangle \;=\; 0 \qquad \text{for every } \varepsilon \in \{-1,1\}^{n-1}. \tag{39}$$

Now fix any $k \neq i$ and choose $\varepsilon, \varepsilon' \in \{-1,1\}^{n-1}$ that agree in every coordinate except the $k$-th, with $\varepsilon_k = 1$ and $\varepsilon'_k = -1$. Subtracting the two instances of (39) gives

$$0 \;=\; \langle v, \varepsilon \rangle - \langle v, \varepsilon' \rangle \;=\; 2\, J_{ik},$$

so $J_{ik} = 0$. Since $k \neq i$ was arbitrary and $J_{ii} = 0$ by hypothesis, the $i$-th row of $J$ vanishes. As $i$ was arbitrary, $J = 0$. $\square$

*Remark B.2.* The result directly implies that the set of one-flip local minima $\mathrm{loc}(\mathcal{E})$ is strictly smeller than the entire set $\{-1,1\}^n$ for all nonzero Ising model ($\mathbf{J} \neq \mathbf{0}$). Negation of the statement means: for all $\mathbf{J} \neq \mathbf{0}$, there exists $s \in \{-1,1\}^n$ and index $i$ such that $s_i(\mathbf{J}s)_i < 0$, implying the existence of $\gamma$ in (19).

**Lemma B.3.** *Let $f \colon \mathbb{R}^n \to \mathbb{R}$ be differentiable and let $x \in \mathbb{D}$ be a vertex of the cube $C = [-\lambda, \lambda]^n$.*

(a) **(Necessary)** *If $x$ is a local minimum of $f$ over $C$, then*

$$x_i \frac{\partial f}{\partial x_i}(x) \leqslant 0 \qquad \forall\, i = 1, \dots, n. \tag{40}$$

(b) **(Sufficient)** *If the strict inequalities*

$$x_i \frac{\partial f}{\partial x_i}(x) < 0 \qquad \forall\, i = 1, \dots, n \tag{41}$$

*hold, then $x$ is a strict local minimum of $f$ over $C$.*

*Proof.* Since every constraint $-\lambda \leqslant x_i \leqslant \lambda$ is active at a vertex, the cone of feasible directions at $x$ is

$$\mathcal{F}_x \;=\; \{\, d \in \mathbb{R}^n : x_i\, d_i \leqslant 0 \ \forall i \,\}.$$

**(a) Necessary condition.** Fix $i \in \{1, \dots, n\}$ and set $d = -x_i\, e_i/\lambda$, where $e_i$ is the $i$-th standard basis vector. Then $x_i d_i = -x_i^2/\lambda = -\lambda < 0$, so $d \in \mathcal{F}_x$ and $x + td \in C$ for all sufficiently small $t > 0$. Because $x$ is a local minimum,

$$0 \;\leqslant\; \lim_{t \downarrow 0} \frac{f(x + td) - f(x)}{t} \;=\; \nabla f(x)^\top d \;=\; -\frac{x_i}{\lambda} \frac{\partial f}{\partial x_i}(x),$$

which gives $x_i \frac{\partial f}{\partial x_i}(x) \leqslant 0$.

**(b) Sufficient condition.** Assume (41) holds. Let $S = \{d \in \mathcal{F}_x : \|d\| = 1\}$. For every $d \in S$ and every index $i$, the factor $\frac{\partial f}{\partial x_i}(x)$ has the opposite sign to $x_i$ (by (41)), while $d_i$ also has the opposite sign to $x_i$ or is zero (by feasibility); hence each product $\frac{\partial f}{\partial x_i}(x)\, d_i \geqslant 0$, with strict inequality for at least one index (since $\|d\| = 1$). Therefore $\nabla f(x)^\top d > 0$ on the compact set $S$, and

$$\delta := \min_{d \in S} \nabla f(x)^\top d > 0.$$

Now take any $y \in C \setminus \{x\}$ and write $r = \|y - x\|$. By differentiability,

$$f(y) - f(x) = r \cdot \nabla f(x)^\top (y - x)/r + o(r) \geqslant r\,\delta + o(r).$$

For $r > 0$ small enough, $r\,\delta + o(r) > 0$, so $f(y) > f(x)$. $\qquad\square$

## B.1. Proof of Proposition 3.4

**Re-statement:** Let matrix $\mathbf{J}$ be stored with finite precision such that $\mathrm{lsb}(\mathbf{J}) = 10^{-b}$, where $b \in \mathbb{Z}$ (lowest significant bit in base 10). Then,

$$\gamma(\mathbf{J}) \geqslant \mathrm{lsb}(\mathbf{J}). \tag{42}$$

*Proof.* First, consider the case where $\mathbf{J}$ consists entirely of integers, i.e., $\mathbf{J} \in \mathbb{Z}^{n \times n}$. For any spin configuration $s \in \{-1, 1\}^n$, the local field term $(\mathbf{J}s)_i = \sum_j J_{ij}s_j$ is a sum of integers, and thus must be an integer. If a configuration is violating (i.e., $s_i(\mathbf{J}s)_i < 0$), then $(\mathbf{J}s)_i \neq 0$. Since $(\mathbf{J}s)_i \in \mathbb{Z} \setminus \{0\}$, we have $|(\mathbf{J}s)_i| \geqslant 1$. Consequently, for integer matrices, $\gamma(\mathbf{J}) \geqslant 1$.

Now, consider the general case where $\mathbf{J}$ is stored with precision $10^{-b}$. We can express $\mathbf{J}$ as a scaled integer matrix:

$$\mathbf{J} = 10^{-b} \cdot \mathbf{K}, \quad \text{where } \mathbf{K} \in \mathbb{Z}^{n \times n}.$$

From the definition of $\gamma$ in (19), observe the homogeneity property for any scalar $c > 0$:

$$\begin{aligned}
\gamma(c\mathbf{J}) &= \min_s \left\{ |c(\mathbf{J}s)_i| : s_i c(\mathbf{J}s)_i < 0 \right\} \\
&= c \cdot \min_s \left\{ |(\mathbf{J}s)_i| : s_i(\mathbf{J}s)_i < 0 \right\} \\
&= c\gamma(\mathbf{J}).
\end{aligned}$$

Applying this to our decomposition:

$$\gamma(\mathbf{J}) = \gamma(10^{-b}\mathbf{K}) = 10^{-b}\gamma(\mathbf{K}).$$

Since $\mathbf{K}$ is an integer matrix, $\gamma(\mathbf{K}) \geqslant 1$. Therefore:

$$\gamma(\mathbf{J}) \geqslant 10^{-b} \cdot 1 = \mathrm{lsb}(\mathbf{J}).$$

$\qquad\square$

## B.2. Proof of Lemma 3.7

**Re-statement:** Assuming (21), every local minimizer of (9) lies on the hypercube corners $\mathbb{D} = \{-\lambda, \lambda\}^n$, i.e.

$$\mathrm{loc}(\mathcal{H}) \subseteq \mathbb{D}. \tag{43}$$

*Proof.* Let $x^*$ be a local minimizer of (9) such that $x^* \notin \mathbb{D}$. Then there exists at least one coordinate $i$ with $x_i^* \in (-\lambda, \lambda)$. Since $x^*$ lies in the interior with respect to the $i$-th coordinate, the (unconstrained) second-order necessary condition (since $\mathcal{H}$ is twice-differentiable) for a local minimum implies

$$\frac{\partial^2}{\partial x_i^2}\mathcal{H}(x^*) \geqslant 0.$$

Using $\mathcal{H}(\boldsymbol{x}) = \mathcal{E}(\boldsymbol{x}) + \mathcal{A}_f(\boldsymbol{x})$ with $\mathcal{A}_f(\boldsymbol{x}) = \sum_{j=1}^{n} f_\theta(x_j)$ and $\mathcal{E}(\boldsymbol{x}) = -\frac{1}{2}\boldsymbol{x}^\top \mathbf{J}\boldsymbol{x}$, we have

$$\frac{\partial^2}{\partial x_i^2}\mathcal{H}(\boldsymbol{x}) = \frac{\partial^2}{\partial x_i^2}\mathcal{E}(\boldsymbol{x}) + \frac{\partial^2}{\partial x_i^2}\mathcal{A}_f(\boldsymbol{x}) = -J_{ii} + f_\theta''(x_i).$$

Now for Ising model $J_{ii} = 0$ (no self-coupling) hence,

$$\frac{\partial^2}{\partial x_i^2}\mathcal{H}(\boldsymbol{x}^*) = f_\theta''(x_i^*) < 0,$$

contradicting the local minimality of $\boldsymbol{x}^*$ along the $i$-th direction. Hence, no such $\boldsymbol{x}^*$ can exist, and every local minimizer must satisfy $\boldsymbol{x}^* \in \mathbb{D}$. $\qquad\square$

### B.3. Proof of Lemma 3.9

**Re-statement:** Assuming (20) and (22), every discrete one-flip local minimizer $\boldsymbol{s}^* \in \mathrm{loc}(\mathcal{E})$ induces a local minimizer $\boldsymbol{x}^* = \lambda \boldsymbol{s}^*$ of (9). Equivalently,

$$S_\lambda^* \subseteq \mathrm{loc}(\mathcal{H}). \tag{44}$$

*Proof.* Let $\boldsymbol{x} \in S_\lambda^*$, so $\boldsymbol{x} = \lambda \boldsymbol{s}$ for some $\boldsymbol{s} \in \mathrm{loc}(\mathcal{E})$ and hence $\mathrm{sign}(\boldsymbol{x}) = \boldsymbol{s}$. From (17), $\boldsymbol{s} \in \mathrm{loc}(\mathcal{E})$ implies

$$s_i(\mathbf{J}\boldsymbol{s})_i \geqslant 0, \qquad \forall i. \tag{45}$$

Multiplying by $\lambda^2$ and using $\boldsymbol{x} = \lambda \boldsymbol{s}$ gives

$$x_i(\mathbf{J}\boldsymbol{x})_i \geqslant 0, \qquad \forall i. \tag{46}$$

Now consider the constrained first-order optimality condition for $\boldsymbol{x} \in [-\lambda, \lambda]^n$. From Lemma B.3, a corner point $x \in \mathbb{D}$ will be a strict local minimizer if

$$x_i \frac{\partial}{\partial x_i}\mathcal{H}(\boldsymbol{x}) < 0, \qquad \forall i. \tag{47}$$

Since $\mathcal{H}(\boldsymbol{x}) = \mathcal{E}(\boldsymbol{x}) + \sum_{j=1}^{n} f_\theta(x_j)$ and $\nabla\mathcal{E}(\boldsymbol{x}) = -\mathbf{J}\boldsymbol{x}$, we have

$$\frac{\partial}{\partial x_i}\mathcal{H}(\boldsymbol{x}) = -(\mathbf{J}\boldsymbol{x})_i + f_\theta'(x_i).$$

Thus,

$$\begin{aligned}
x_i \frac{\partial}{\partial x_i}\mathcal{H}(\boldsymbol{x}) &= x_i\big(-(\mathbf{J}\boldsymbol{x})_i + f_\theta'(x_i)\big) \\
&= -x_i(\mathbf{J}\boldsymbol{x})_i + x_i f_\theta'(x_i) \\
&\leqslant x_i f_\theta'(x_i),
\end{aligned}$$

where we used (46). Since $f_\theta$ is even (from (20)) and differentiable, $f_\theta'$ is odd, and therefore

$$x_i f_\theta'(x_i) = \lambda f_\theta'(\lambda) < 0, \qquad \forall x_i \in \{\pm\lambda\},$$

by (22). Hence (47) holds for all $i$, proving that $\boldsymbol{x}$ is a local minimizer of (9). $\qquad\square$

### B.4. Proof of Lemma 3.11

**Re-statement:** Assuming (20), (21) and (23), every local minimizer $\boldsymbol{x}^* \in \mathrm{loc}(\mathcal{H})$ induces a discrete one-flip local minimum $\mathrm{sign}(\boldsymbol{x}^*) \in \mathrm{loc}(\mathcal{E})$, i.e.

$$\mathrm{loc}(\mathcal{H}) \subseteq S_\lambda^*. \tag{48}$$

*Proof.* Let $\boldsymbol{x} \in \mathrm{loc}(\mathcal{H})$. By (21) Lemma 3.7 holds so, $\boldsymbol{x} \in \mathbb{D}$, and $x_i \in \{\pm\lambda\}$ for all $i$. Let $\boldsymbol{s} := \mathrm{sign}(\boldsymbol{x}) \in \{-1, 1\}^n$, so $\boldsymbol{x} = \lambda \boldsymbol{s}$.

Assume, for contradiction, that $s \notin \text{loc}(\mathcal{E})$. Then by definition of $\text{loc}(\mathcal{E})$ and (17), there exists an index $i$ such that

$$s_i(\mathbf{J}s)_i < 0. \tag{49}$$

By the definition of $\gamma$ in (19), this implies

$$|(\mathbf{J}s)_i| \geqslant \gamma. \tag{50}$$

Multiplying (50) by $-\lambda^2$ and using $\boldsymbol{x} = \lambda \boldsymbol{s}$ yields

$$x_i(\mathbf{J}\boldsymbol{x})_i = \lambda^2 s_i(\mathbf{J}s)_i \leqslant -\lambda^2 \gamma. \tag{51}$$

Since $\boldsymbol{x}$ is a constrained local minimizer on $[-\lambda, \lambda]^n$ and lies on the boundary in every coordinate, from Lemma B.3 it must satisfy the first-order condition:

$$x_i \frac{\partial}{\partial x_i} \mathcal{H}(\boldsymbol{x}) \leqslant 0.$$

Using $\partial_{x_i} \mathcal{H}(\boldsymbol{x}) = -(\mathbf{J}\boldsymbol{x})_i + f'_\theta(x_i)$, we obtain

$$\begin{aligned}
0 &\geqslant x_i \frac{\partial}{\partial x_i} \mathcal{H}(\boldsymbol{x}) \\
&= x_i \big( -(\mathbf{J}\boldsymbol{x})_i + f'_\theta(x_i) \big) \\
&= -x_i(\mathbf{J}\boldsymbol{x})_i + x_i f'_\theta(x_i) \\
&\geqslant \lambda^2 \gamma + x_i f'_\theta(x_i), \qquad \text{using (51)}.
\end{aligned}$$

Since $x_i \in \{\pm\lambda\}$ and $f'_\theta$ is odd (from (20)), we have $x_i f'_\theta(x_i) = \lambda f'_\theta(\lambda)$. Hence

$$0 \geqslant \lambda^2 \gamma + \lambda f'_\theta(\lambda) \quad \Longleftrightarrow \quad f'_\theta(\lambda) + \lambda\gamma \leqslant 0,$$

which contradicts (23). Therefore $\boldsymbol{s} \in \text{loc}(\mathcal{E})$, and hence $\boldsymbol{x} \in S^*_\lambda$. $\qquad \square$

### B.5. Proof of Theorem 3.13

**Re-statement:** Assuming $f_\theta$ is admissible, i.e. it satisfies (20) to (23), $\boldsymbol{x}^* \in \text{loc}(\mathcal{H})$ if and only if $\text{sign}(\boldsymbol{x}^*) \in \text{loc}(\mathcal{E})$.

*Proof.* The assumptions (22) and (23) together imply two distinct inequalities required by the supporting lemmas.

Assumption (20) and (22) satisfies the condition of Lemma 3.9, ensuring that every discrete minimum induces a continuous minimum:

$$S^*_\lambda \subseteq \text{loc}(\mathcal{H}).$$

Assumption (20) and (23) satisfies the conditions of Lemma 3.11, ensuring that every continuous minimum induces a discrete minimum:

$$\text{loc}(\mathcal{H}) \subseteq S^*_\lambda.$$

Combining both inclusions yields $\text{loc}(\mathcal{H}) = S^*_\lambda$. By definition of $S^*_\lambda$, this establishes the equivalence:

$$\boldsymbol{x}^* \in \text{loc}(\mathcal{H}) \iff \text{sign}(\boldsymbol{x}^*) \in \text{loc}(\mathcal{E}).$$

$\qquad \square$

## C. MAX-CUT and GW-SDP

In this section, we provide the detailed reformulation of the MAX-CUT problem into the Ising framework and outline the Goemans-Williamson Semidefinite Programming (GW-SDP) relaxation (Goemans & Williamson, 1995), which serves as a standard benchmark for approximation quality.

We consider an undirected weighted graph $G = (V, E)$ with a weighted adjacency matrix $\mathbf{W} \in \mathbb{R}^{n \times n}$, where $W_{ij} = W_{ji}$ represents the weight of the edge between vertices $i$ and $j$, and $W_{ij} = 0$ if no edge exists. We seek a partition of $V$ into two disjoint sets $A$ and $B$ to maximize the total weight of edges crossing the partition.

### C.1. Ising Model Formulation

Using the spin encoding $s \in \{-1, 1\}^n$, we assign $s_i = +1$ if $i \in A$ and $s_i = -1$ if $i \in B$. The indicator function for an edge $(i, j)$ being in the cut (i.e., $\mathbb{1}_{\{s_i \neq s_j\}}$) can be written algebraically as $\frac{1 - s_i s_j}{2}$. Consequently, the cut value in (3) becomes:

$$
\begin{aligned}
\text{Cut}(s) &= \sum_{(i,j) \in E} W_{ij} \left( \frac{1 - s_i s_j}{2} \right) \\
&= \frac{1}{2} \sum_{(i,j) \in E} W_{ij} - \frac{1}{2} \sum_{(i,j) \in E} W_{ij} s_i s_j \\
&= \text{constant} - \frac{1}{4} s^\top \mathbf{W} s.
\end{aligned}
\tag{52}
$$

Maximizing the cut value is therefore equivalent to minimizing the quadratic form $s^\top \mathbf{W} s$. By defining the interaction matrix as $\mathbf{J} = -(1/2)\mathbf{W}$ (as established in Section A), the optimization problem becomes equivalent to finding the ground state of the Ising energy:

$$
\max_s \text{Cut}(s) \iff \min_s \frac{1}{4} s^\top \mathbf{W} s \iff \min_s -\frac{1}{2} s^\top \mathbf{J} s.
\tag{53}
$$

### C.2. Semidefinite Relaxation

The GW-SDP relaxation relies on lifting the vector variable $s$ into a matrix space. We define the matrix variable $\mathbf{X} = ss^\top$. The objective function can then be rewritten using the matrix trace operator:

$$
s^\top \mathbf{W} s = \text{Trace}(\mathbf{W} ss^\top) = \text{Trace}(\mathbf{W}\mathbf{X}).
\tag{54}
$$

The constraint $s \in \{-1, 1\}^n$ implies that $\mathbf{X}$ must be a rank-one positive semidefinite matrix ($\mathbf{X} \succeq \mathbf{0}$) with diagonal elements $X_{ii} = s_i^2 = 1$. The non-convexity of the original problem is entirely captured by the rank-one constraint. By dropping this constraint, we obtain the convex Semidefinite Program (SDP):

$$
\begin{aligned}
\underset{\mathbf{X}}{\text{minimize}} \quad & \text{Trace}(\mathbf{W}\mathbf{X}) \\
\text{subject to} \quad & \mathbf{X} \succeq \mathbf{0}, \\
& X_{ii} = 1, \quad \forall i = 1, \ldots, n.
\end{aligned}
\tag{55}
$$

This relaxation can be solved in polynomial time using interior-point methods. However, the computational cost scales as $\mathcal{O}(n^{4.5})$ or $\mathcal{O}(n^{3.5})$ depending on the solver implementation (Luo et al., 2010), which restricts its scalability to graphs with approximately $10^3$ to $10^4$ vertices.

Once the optimal matrix $\mathbf{X}^*$ is obtained from (55), a valid discrete spin configuration $s$ is recovered via a randomized rounding scheme (Gaussian rounding). Goemans and Williamson proved that the expected cut value is at least $87.8\%$ of the optimal MAC-CUT value.

## D. Physics-Inspired Class of Attractor Functions

### D.1. Connection to Nonlinear Parametric Oscillators

In this section, we discuss the physical motivation behind our choice of attractor functions. Specifically, we consider the general class of polynomial attractors $\mathcal{A}_f(x) = \sum_{i=1}^n f_\theta(x_i)$, where the scalar potential $f_\theta$ is defined as:

$$
f_\theta(x) = \frac{\beta}{2k+2} x^{2k+2} - \frac{\alpha}{2k} x^{2k}, \quad \theta = (\alpha, \beta, k) > 0, \, k \in \mathbb{N},
\tag{56}
$$

We show that for the specific case of $k = 1$, the resulting Hamiltonian corresponds directly to the effective Hamiltonian of the Kerr-nonlinear parametric oscillator (KPO) networks (Goto, 2018).

The Hamiltonian for a network of KPOs is generally given by:

$$
\mathcal{H}_c(x, y, t) = \sum_{i=1}^N \left( \frac{K}{4} (x_i^2 + y_i^2)^2 - \frac{p(t)}{2} (x_i^2 - y_i^2) + \frac{\Delta_i}{2} (x_i^2 + y_i^2) \right) - \frac{\xi_0}{2} \sum_{i=1}^N \sum_{j=1}^N J_{ij} (x_i x_j + y_i y_j),
\tag{57}
$$

where $\xi_0$ is a positive coupling constant, $K$ is the Kerr nonlinearity coefficient, $p(t)$ is the time-dependent parametric pumping amplitude, $\Delta_i$ is the detuning frequency of the $i$-th oscillator, $x_i$ represents the in-phase amplitude (analogous to the spin state), and $y_i$ represents the quadrature amplitude (analogous to momentum).

By considering the simplified static regime where the quadrature components are suppressed ($y_i = 0$) and assuming uniform detuning ($\Delta_i = \Delta$), Equation (57) reduces to:

$$\mathcal{H}_{KPO}(\boldsymbol{x}, \mathbf{0}, t) = \sum_{i=1}^{N} \left( \frac{K}{4}x_i^4 - \frac{p(t) - \Delta}{2}x_i^2 \right) - \frac{\xi_0}{2} \sum_{i=1}^{N} \sum_{j=1}^{N} J_{ij} x_i x_j$$

$$= \sum_{i=1}^{N} \left( \underbrace{\frac{K}{4}x_i^4 - \frac{p(t) - \Delta}{2}x_i^2}_{f_\theta(x_i)} \right) - \frac{\xi_0}{2} \boldsymbol{x}^\top \mathbf{J} \boldsymbol{x}. \tag{58}$$

Comparing the inner sum to the class of attractor function (56) directly yields

$$\alpha = p(t) - \Delta, \quad \beta = K, \quad k = 1.$$

### D.2. Connection to Mean-Field Annealing on Ising Model

In this section, we demonstrate that the attractor defined by:

$$f_\theta(x) = \frac{1}{\beta} \int_0^x \tanh^{-1}\left(\frac{t}{\lambda}\right) dt - \frac{\alpha}{2}x^2, \quad \theta = (\alpha, \beta) > 0, \tag{59}$$

naturally arises from the mean-field approximation of the Ising model, establishing a rigorous link between our continuous relaxation and statistical mechanics approaches.

Consider the thermal equilibrium of the Ising model described by the Gibbs distribution $P(\boldsymbol{s}) \propto \exp(-\beta \mathcal{E}(\boldsymbol{s}))$, where $\beta$ represents the inverse temperature. In the mean-field approximation, spins are assumed to fluctuate independently in an effective local field created by their neighbours. The expectation value of the $i$-th spin, denoted by $x_i = \langle s_i \rangle \in \{-\lambda, \lambda\}$, satisfies the well-known self-consistency equation (King et al., 2018):

$$x_i = \lambda \tanh \left( h_i x_i + \beta \sum_j J_{ij} x_j \right). \tag{60}$$

Solving the combinatorial problem via Mean-Field Annealing (MFA) typically involves iterating this fixed-point equation while gradually increasing $\beta$ (lowering temperature).

We now examine the stationary points of our Hamiltonian $\mathcal{H}(\boldsymbol{x}) = \mathcal{E}(\boldsymbol{x}) + \mathcal{A}_f(\boldsymbol{x})$ using the attractor in (59). The gradient condition for a stationary point is $\nabla \mathcal{H}(\boldsymbol{x}) = \mathbf{0}$. Setting the total gradient to zero yields:

$$-\mathbf{J}\boldsymbol{x} + \frac{1}{\beta}\tanh^{-1}\left(\frac{\boldsymbol{x}}{\lambda}\right) - \alpha\boldsymbol{x} = \mathbf{0}$$

$$\frac{1}{\beta}\tanh^{-1}\left(\frac{\boldsymbol{x}}{\lambda}\right) = (\mathbf{J} + \alpha\mathbf{I})\boldsymbol{x}. \tag{61}$$

Rearranging for $\boldsymbol{x}$, we apply the hyperbolic tangent function to both sides:

$$\boldsymbol{x} = \lambda \tanh\left(\beta(\mathbf{J} + \alpha\mathbf{I})\boldsymbol{x}\right). \tag{62}$$

Thus, the stationary points of our continuously relaxed objective with the attractor (59) correspond exactly to the fixed points of the Mean-Field Annealing equations, with the parameter $\alpha$ acting as a constant external magnetic field equal to $h/\beta$.

## E. Adaptive Hyperparameter Tuning for MiP-CRIM

The performance of our continuous Ising solvers depends on the shape of the energy landscape, controlled by the parameters $\alpha$ and $\beta$ of the quartic attractor $f_{\theta2}$ in (13), as well as the optimization learning rate $\tau$ (for ADAM) and box radius $\lambda$ in (9). To ensure we operate within the theoretically valid regime (Remark 3.14) while maximizing solution quality, we employ a multi-resolution adaptive grid search.

**Admissible Search Space.** For the polynomial attractor $f_\theta(x) = \frac{\beta}{4}x^4 - \frac{\alpha}{2}x^2$, admissibility (as per Definition 3.6 and Remark 3.14) requires:

$$3\beta\lambda^2 < \alpha < \beta\lambda^2 + \gamma. \tag{63}$$

Rather than searching for $\alpha$ and $\lambda$ independently, we fix a baseline $\gamma_0$ (as per Proposition 3.4 and Remark 3.5) and sweep over $\beta$ with $\gamma \geqslant \gamma_0$. This allows us to derive the dependent parameters:

$$\lambda = \sqrt{\frac{\gamma}{2\beta}}, \quad \alpha \in (3\beta\lambda^2, \beta\lambda^2 + \gamma).$$

This parameterization guarantees that every sampled configuration satisfies the advisability conditions (20)–(23) of our theory. We fix the exponential decay rate parameters for ADAM to 0.09 for the 1st moment, 0.999 for the 2nd moment for all experiments. We choose the noise scaling $\sigma$ between $10^{-3}$ and $10^{-1}$ based on the problem scale.

**Adaptive Refinement Strategy.** We use a "zoom-in" strategy over $R$ rounds to fine-tune the grid:

1. **Initialization:** We define initial ranges for the learning rate $\tau \in (0, \tau_0]$, stiffness $\beta \in (0, \beta_0]$, and effective margin lower bound $\gamma_0 > 0$.

2. **Coarse Sweep:** In each round, we discretize the current interval into a small coarse grid (3 to 5 points along each dimension for efficiency) and run a short solver pass at each grid point.

3. **Selection:** We select the configuration that yields the lowest energy state.

4. **Refinement:** The search range for the next round is centered around the best parameters from the current round, with the window size reduced by half (or doubled if the optimal value lies on the boundary).

This approach, implemented on GPU, allows us to rapidly converge to instance-specific optimal hyperparameters (typically within seconds), balancing the exploration-exploitation trade-off required for challenging NP-hard benchmarks.

## F. Additional Experiments

### F.1. MAX-CUT Results on G-set graphs

We evaluated MiP-CRIM on the standard G-set benchmark using the adaptive strategy described above. For these experiments, we fix the search hyperparameters to $R = 10$ refinement rounds, with each round consisting of $K = 200$ epochs and $T = 10$ ADAM iterations (per epoch). The noise scaling is set to $\sigma = 10^{-3}$. The resulting instance-specific hyperparameters $(\alpha, \beta, \lambda, \gamma)$ for the polynomial attractor function $f_{\theta 2}$, along with the computed cut values and the best-known cut-values (from (Ma & Hao, 2017)) are reported in Table 4 below.

*Table 4.* Parameter settings and results of MiP-CRIM for MAX-CUT on G-set graphs. Best-known values are taken from (Ma & Hao, 2017). The results are obtained using NVIDIA L40S GPU with Intel Xeon 6520P CPU.

| G-set | Nodes | $\alpha$ | $\beta$ | $\lambda$ | $\gamma$ | Best-Known | Cut Value | sync |
|-------|-------|----------|---------|-----------|----------|------------|-----------|------|
| G1 | 800 | 0.0094 | 5.00 | 0.0250 | 0.0063 | 11624 | 11609 | 797 |
| G2 | 800 | 0.0563 | 10.00 | 0.0306 | 0.0750 | 11620 | 11606 | 795 |
| G3 | 800 | 0.0211 | 2.50 | 0.0306 | 0.0188 | 11622 | 11622 | 797 |
| G4 | 800 | 0.0750 | 5.00 | 0.0707 | 0.2000 | 11646 | 11578 | 799 |
| G10 | 800 | 0.0281 | 3.75 | 0.0354 | 0.0375 | 2000 | 2000 | 797 |
| G11 | 800 | 0.0750 | 5.00 | 0.0500 | 0.1000 | 564 | 560 | 737 |
| G12 | 800 | 0.0375 | 3.75 | 0.0408 | 0.0500 | 556 | 554 | 729 |
| G13 | 800 | 0.1500 | 5.00 | 0.1000 | 0.1000 | 582 | 574 | 737 |
| G14 | 800 | 0.0281 | 3.75 | 0.0354 | 0.0375 | 3064 | 3058 | 763 |
| G15 | 800 | 0.0422 | 3.75 | 0.0354 | 0.0375 | 3050 | 3037 | 792 |
| G16 | 800 | 0.0375 | 7.50 | 0.0408 | 0.1000 | 3052 | 3045 | 757 |
| G17 | 800 | 0.0563 | 3.75 | 0.0500 | 0.0750 | 3047 | 3036 | 770 |
| G18 | 800 | 0.0188 | 2.50 | 0.0500 | 0.0125 | 992 | 985 | 779 |

**Table 4 – continued from previous page**

| G-set | Nodes | $\alpha$ | $\beta$ | $\lambda$ | $\gamma$ | Best-Known | Cut Value | Sync |
|-------|-------|----------|---------|-----------|----------|------------|-----------|------|
| G19 | 800 | 0.1500 | 30.00 | 0.0408 | 0.4000 | 906 | 903 | 765 |
| G20 | 800 | 0.1125 | 40.00 | 0.0306 | 0.0750 | 941 | 937 | 761 |
| G21 | 800 | 0.1125 | 5.00 | 0.0500 | 0.1000 | 931 | 921 | 765 |
| G22 | 2000 | 0.1500 | 5.00 | 0.1000 | 0.1000 | 13359 | 13226 | 1965 |
| G23 | 2000 | 0.0563 | 10.00 | 0.0306 | 0.0750 | 13344 | 13304 | 1973 |
| G24 | 2000 | 0.0070 | 1.25 | 0.0306 | 0.0094 | 13337 | 13294 | 1972 |
| G25 | 2000 | 0.1500 | 10.00 | 0.0500 | 0.2000 | 13340 | 13310 | 1976 |
| G26 | 2000 | 0.0105 | 0.63 | 0.0433 | 0.0094 | 13328 | 13286 | 1971 |
| G27 | 2000 | 0.0375 | 20.00 | 0.0250 | 0.1000 | 3341 | 3300 | 1946 |
| G28 | 2000 | 0.0750 | 20.00 | 0.0250 | 0.1000 | 3298 | 3262 | 1947 |
| G29 | 2000 | 0.0094 | 2.50 | 0.0354 | 0.0250 | 3405 | 3375 | 1965 |
| G30 | 2000 | 0.2250 | 20.00 | 0.0354 | 0.2000 | 3413 | 3389 | 1955 |
| G31 | 2000 | 0.1688 | 20.00 | 0.0306 | 0.1500 | 3310 | 3291 | 1957 |
| G32 | 2000 | 0.0375 | 7.50 | 0.0408 | 0.0250 | 1410 | 1386 | 1820 |
| G33 | 2000 | 0.0750 | 10.00 | 0.0500 | 0.0500 | 1382 | 1356 | 1827 |
| G34 | 2000 | 0.0750 | 10.00 | 0.0500 | 0.0500 | 1384 | 1360 | 1823 |
| G35 | 2000 | 0.0006 | 0.12 | 0.0408 | 0.0016 | 7687 | 7649 | 1897 |
| G36 | 2000 | 0.0750 | 3.75 | 0.0817 | 0.2000 | 7680 | 7622 | 1894 |
| G37 | 2000 | 0.0375 | 5.00 | 0.0500 | 0.1000 | 7691 | 7643 | 1898 |
| G38 | 2000 | 0.1125 | 1.88 | 0.0817 | 0.1000 | 7688 | 7626 | 1908 |
| G39 | 2000 | 0.0750 | 5.00 | 0.0500 | 0.1000 | 2408 | 2350 | 1881 |
| G40 | 2000 | 0.0422 | 3.75 | 0.0354 | 0.0375 | 2400 | 2358 | 1905 |
| G41 | 2000 | 0.0094 | 1.88 | 0.0408 | 0.0250 | 2405 | 2372 | 1866 |
| G42 | 2000 | 0.0281 | 10.00 | 0.0306 | 0.0188 | 2481 | 2446 | 1897 |
| G43 | 1000 | 0.2250 | 15.00 | 0.0408 | 0.2000 | 6660 | 6656 | 992 |
| G44 | 1000 | 0.1500 | 60.00 | 0.0289 | 0.1000 | 6650 | 6645 | 984 |
| G45 | 1000 | 0.0094 | 5.00 | 0.0250 | 0.0250 | 6654 | 6643 | 984 |
| G46 | 1000 | 0.0070 | 0.31 | 0.0866 | 0.0188 | 6649 | 6620 | 994 |
| G47 | 1000 | 0.2250 | 7.50 | 0.0577 | 0.2000 | 6657 | 6625 | 992 |
| G48 | 3000 | 0.0375 | 5.00 | 0.0500 | 0.1000 | 6000 | 6000 | 3000 |
| G49 | 3000 | 0.0563 | 10.00 | 0.0250 | 0.0500 | 6000 | 6000 | 3000 |
| G50 | 3000 | 0.0375 | 20.00 | 0.0250 | 0.0250 | 5880 | 5856 | 2958 |
| G51 | 1000 | 0.1500 | 3.75 | 0.0817 | 0.2000 | 3848 | 3817 | 956 |
| G52 | 1000 | 0.0422 | 2.50 | 0.0433 | 0.0375 | 3851 | 3838 | 957 |
| G53 | 1000 | 0.0375 | 3.75 | 0.0408 | 0.0500 | 3850 | 3838 | 957 |
| G54 | 1000 | 0.0422 | 1.88 | 0.0500 | 0.0375 | 3852 | 3840 | 955 |
| G55 | 5000 | 0.1125 | 5.00 | 0.0500 | 0.1000 | 10299 | 10173 | 4973 |
| G70 | 10000 | 0.300 | 60.00 | 0.040825 | 0.0200 | 9591 | 9531 | 9949 |

## F.2. Ablation Study

To assess the robustness of MiP-CRIM and validate our adaptive tuning strategy, we conducted an ablation study analyzing the sensitivity of the solver to its three primary hyperparameters: the polynomial attractor coefficient $\alpha$, the box constraint radius $\lambda$, and the optimization learning rate $\tau$. We evaluated performance on three representative benchmarks: the SK Model (1000 spins), the dense K2000 graph, and the sparse G-set graph G10. For each parameter sweep, we report both the solution quality (Ising Energy or Cut Value) and the synchronization score (sync) from (29), which measures the alignment of the continuous variables with the discrete one-flip local minima.

**Sensitivity to Attractor Shape ($\alpha$)**    The parameter $\alpha$ controls the convexity near the origin and the depth of the discrete wells 1. We swept $\alpha$ within the theoretically valid region $(3\beta\lambda^2, \beta\lambda^2 + \gamma)$ derived in our theoretical analysis. As shown in Figure 5, MiP-CRIM maintains a high synchronization score (sync > 0.9) across the entire valid range, confirming that the solver consistently converges to valid discrete states. However, solution quality exhibits a clear trend: lower values of $\alpha$ (closer to the point $3\beta\lambda^2$) generally yield better energies and cut values. This suggests that a "flatter" landscape near the origin facilitates better exploration of the configuration space before the system settles into a specific basin of attraction.

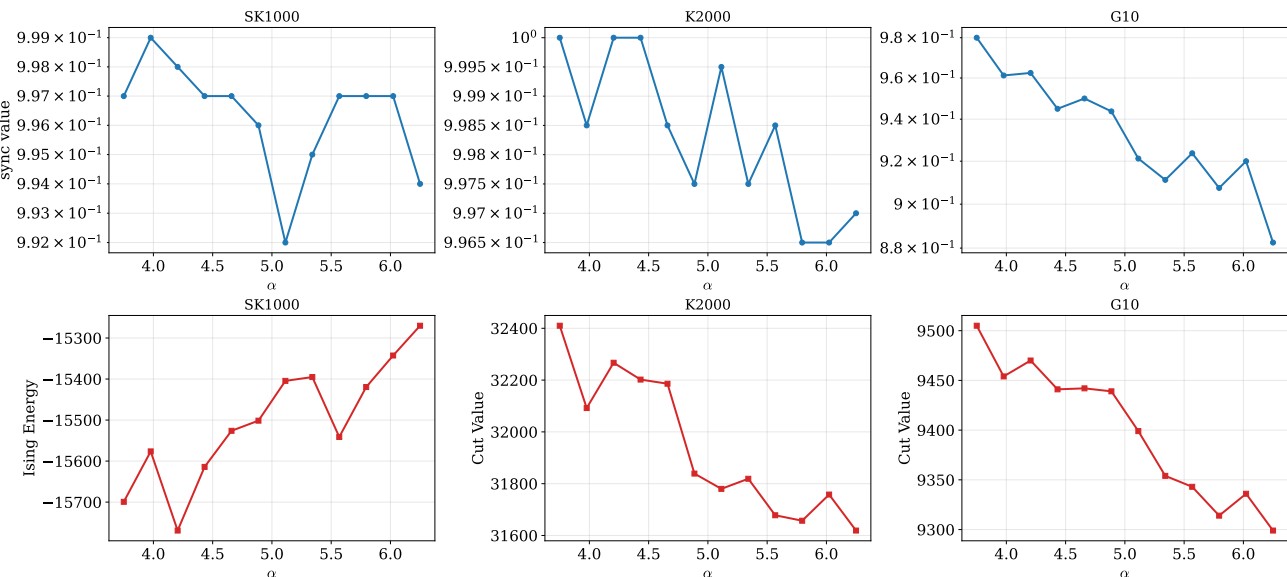

*Figure 5. Sensitivity to Attractor Shape ($\alpha$). Top Row: synchronization score (sync) vs. $\alpha$. Bottom Row: Objective value (Ising Energy for SK1000; Cut Value for K2000 / G10). The x-axis represents the valid theoretical range for $\alpha$. Lower $\alpha$ values promote better exploration (higher cuts/lower energy) while maintaining high synchronization score. The results are obtained using NVIDIA L40S GPU with Intel Xeon 6520P CPU.*

**Sensitivity to Hypercube Length** ($\lambda$)  The parameter $\lambda$ defines the bounds of the continuous relaxation hypercube $[-\lambda, \lambda]^N$ in (9). We evaluated the solver's performance across a logarithmic scale $\lambda \in [10^{-3}, 10^3]$. Figure 6 reveals a distinct "sweet spot" for performance. For all three datasets, optimal solution quality and perfect synchronization score (sync $\approx 1.0$) are achieved for $\lambda \in [10^{-1}, 10^0]$.

- **Small** $\lambda$ ($< 10^{-1}$)**:** The solver remains stable (sync $\approx 1.0$) but performance degrades slightly, likely due to the constrained space limiting the separation between variables.

- **Large** $\lambda$ ($> 10^1$)**:** Performance collapses. The synchronization score drops to $\approx 0.5$ (equivalent to random guessing), and cut values tend toward zero. In this regime, the box constraint becomes loose relative to the polynomial attractor, causing gradients to vanish or the system to drift without bifurcating properly into discrete spins.

**Sensitivity to Learning Rate** ($\tau$).  Finally, we analyzed the impact of the ADAM optimizer's step size $\tau \in [0.05, 0.95]$. Figure 7 demonstrates the remarkable stability of the proposed attractor mechanism to variations in optimization speed. While the synchronization score exhibits minor fluctuations (top row), it remains consistently near-perfect (maintaining values $> 0.99$ for SK1000 and K2000, and $> 0.98$ for G10) across the entire tested range. This indicates that the deterministic mapping to one-flip local optimal states is robust. In contrast, the final Ising energy or cut value (bottom row) varies significantly with $\tau$. This persistent variability justifies the necessity of the multi-resolution grid search employed in our main experiments (Section E) to locate the precise learning rate that maximizes solution quality for a specific instance.

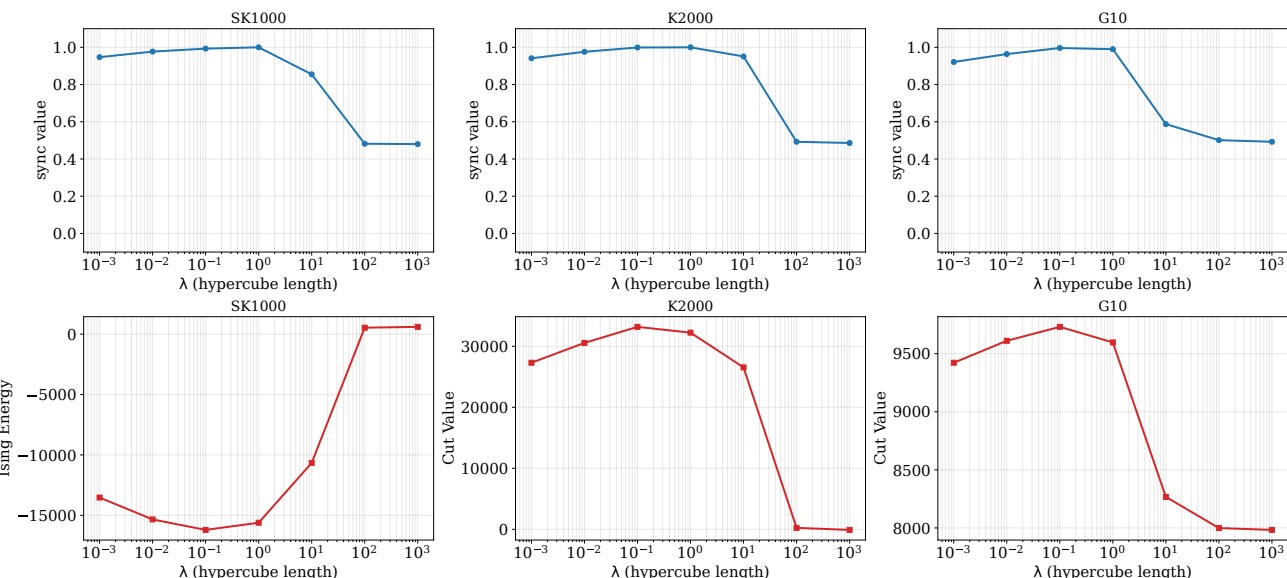

*Figure 6. Sensitivity to Domain Size (λ).* Performance across varying box radii (log scale). A clear operational window exists for $\lambda \in [0.1, 1.0]$ where the solver achieves both optimal objective values and perfect synchronization score. Large $\lambda$ values lead to a collapse in performance. The results are obtained using NVIDIA L40S GPU with Intel Xeon 6520P CPU.

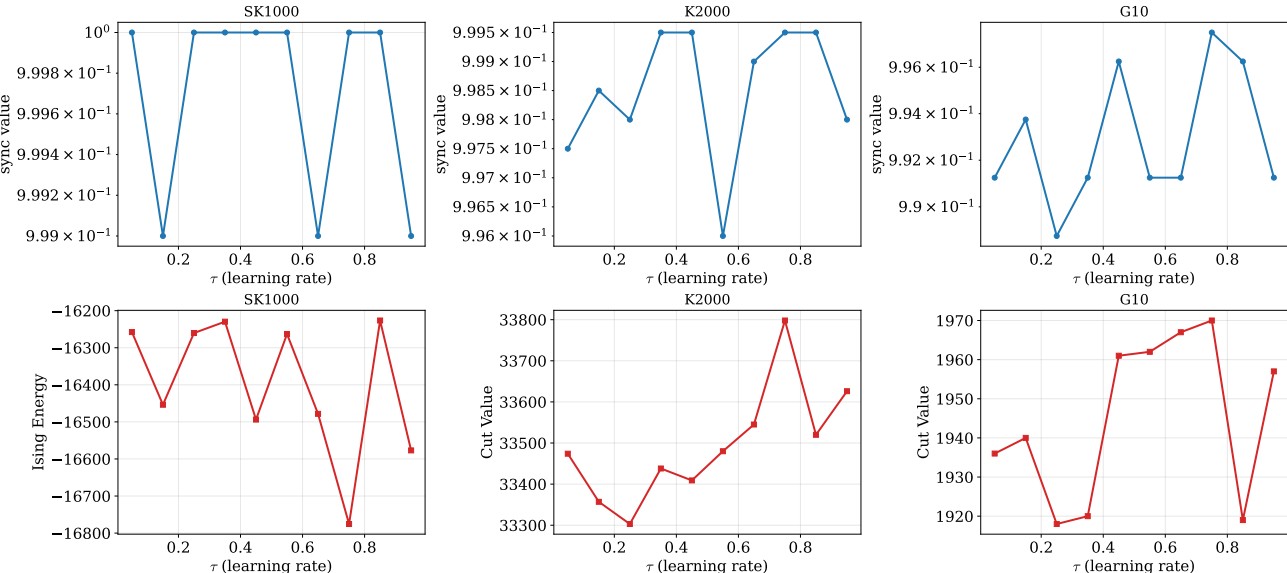

*Figure 7. Sensitivity to Learning Rate (τ).* The synchronization score (top row) is invariant to the learning rate, remaining at 1.0, which highlights the robustness of the MiP-CRIM attractor. Objective values (bottom row) fluctuate, necessitating the adaptive tuning strategy employed in our main experiments. The results are obtained using NVIDIA L40S GPU with Intel Xeon 6520P CPU.

