# OpenReview forum: "Local-Minima-Preserving Polynomial Relaxation of Ising Problems"
_ICML.cc/2026/Conference — ICML 2026 regular_

### Official Review · Reviewer_4ZdA · 2026-03-11

**Soundness:** 3
**Presentation:** 3
**Significance:** 3
**Originality:** 3
**Overall Recommendation:** 4
**Confidence:** 3

**Summary:**

This work introduces the generalized Ising problem, a combinatorial optimization objective which captures MAX-CUT, the number partition problem and maximizing the Hamiltonian of the Sherrington-Kirkpatrick model among others. The authors analyze the problem of finding one-flip local minima of this objective using first order-methods, achieving an algorithm that is significantly faster than existing methods in the literature.

**Compliance With Llm Reviewing Policy:**

Affirmed.

**Final Justification:**

I would like to point out that local optima is the right notion of a solution concept within this area, which even for simple objectives is a difficult task as evidenced by my comment about PLS vs CLS hardness.

The fact that this paper is algorithmic in nature frames the context around the primary result, that is there is no worst-case convergence guarantees of the first-order optimization algorithm, but rather the landscape equivalence between the two problems. For a particular "hard" instance of the problem, it will also probably be difficult for the first-order algorithm to solve it, leading to no contradiction.

With this in mind, I will raise my score in accordance with the strong experimental results over multiple settings of interest.

**Key Questions For Authors:**

>- How does this method scale to arbitrary (non-quadratic) objectives? Does the logic behind the regularizer need to be modified? Does cycling behavior begin to develop?

**Limitations:**

yes

**Strengths And Weaknesses:**

**Strengths**
>- This paper is well-written and easy to follow. The related work is sufficiently discussed and the primary contributions are clear.
>- The novel generalized Ising problem is amply motivated, with applications to classical problems such as MAX-CUT and NPP, and of interest to the optimization community.
>- The battery of experiments is thorough and demonstrates the superiority of their methods on a variety of NP-Hard combinatorial optimization problems. Ablation studies found in the appendix give a comparison between hyperparameters.

**Weaknesses**
>- The paper would benefit from discussing the theoretical limitations of the work. Namely, the problem of finding a local minimum of MAX-CUT or NPP is well-known to be PLS-Complete and on the surface it seems the authors have leveraged the stationary points first-order continuous methods to reformulate the problem, which is a problem within CLS, which is known to be a strict subset of PLS, yielding an apparent contradiction. To avoid this discrepancy, the authors use a key lemma from [1], which posits that first order methods avoid saddle points almost surely, implying that gradient descent when run on this objective will converge to a minima. However, what this analysis leaves out, is that this rate of this convergence could be exponential or even worse making it difficult to provide theoretical convergence guarantees, as it is well known that finding a local minima of a non-convex continuous function is NP-HARD.


Panageas, Ioannis, Georgios Piliouras, and Xiao Wang. "First-order methods almost always avoid saddle points: The case of vanishing step-sizes." Advances in Neural Information Processing Systems 32 (2019).

---

> ### Author Rebuttal · Authors · 2026-03-27
>
> We are glad that the reviewer finds the paper well-written, clearly motivated, and supported by thorough experiments. We appreciate the insightful comments on the theoretical and complexity aspects, which we address and clarify below.
>
> ---
>
> ### **C1. The paper would benefit from discussing the theoretical limitations of the work. Namely, ... yielding an apparent contradiction. To avoid this discrepancy, the authors use a key lemma from [1], which posits that first order methods avoid saddle points almost surely, implying that gradient descent when run on this objective will converge to a minima.**
>
> The main areas for theoretical improvement would be adding stronger convergence guarantees and extending the method to higher-order (non-quadratic) objectives (discussed under **C2, C3**).
>
> We agree that our work does **not** imply a reduction of a PLS-complete problem to a polynomial-time CLS procedure. Our main result is the **landscape equivalence theorem**:
>
> "Under admissibility conditions **every local minimum corresponds exactly to a one-flip discrete local minimum and vice-versa**"
>
> And we make no claim of solving a PLS-complete problem in polynomial time. We request the reviewer to check our comments to **C1, C2** of the reviewer **HXZ2**, where we explain the effectiveness of our method.
>
> ---
>
> ### **C2. However, what this analysis leaves out, is that this rate of this convergence could be exponential or even worse making it difficult to provide theoretical convergence guarantees, as it is well known that finding a local minima of a non-convex continuous function is NP-HARD.**
>
> We agree that convergence to a local minimum of a nonconvex function can be difficult in the worst case, and we do **not** claim worst-case polynomial-time guarantees.
>
> Our use of first-order methods (e.g., ADAM) is **algorithmic and heuristic**, not a complexity-theoretic claim. The role of our formulation is to construct a landscape where:
>
> * interior stationary points are non-minimizing saddle points
> * local minima correspond exactly to discrete one-flip local minima
>
> The convergence in practice is due to the effectiveness of the ADAM optimizer, which is chosen after doing exhaustive experiments on different gradient and momentum-based optimizers (*Figure 2* of the paper). The convergence is further enhanced by the basin-hopping strategy using Gaussian perturbation (step 15 of *algorithm 1* in the paper), which by [Panageas et al.] ensures our solver escapes the saddle points interior to the domain $[-\lambda, \lambda]^n$ and converges to the discrete local minima (due to the admissibility) at the corner-points.
>
> In short, convergence is aided by:
>
> * random initialization
> * basin-hopping via Gaussian perturbations (step 15 of *Algorithm 1*)
> * efficient first-order dynamics (ADAM)
>
> ---
>
> ### **C3. How does this method scale to arbitrary (non-quadratic) objectives? Does the logic behind the regularizer need to be modified? Does cycling behavior begin to develop?**
>
> Our method is designed for the standard Ising model, which encloses the class of all Quadratic Unconstrained Binary Optimization (QUBO) problems. Here, a single-spin flip admits a local-field characterization that enables our analysis. For general non-quadratic (polynomial) objectives (e.g. PUBO class), this structure does not hold, so the current theory does not directly extend. An extension to the PUBO class will be a clear and meaningful direction for future work.
>
> ---
>
> ### **Summary:**
>
> * No PLS to CLS reduction is claimed.
> * No worst-case guarantees are claimed, but strong empirical convergence is observed.
> * Method is applicable to **generalized Ising model** (covering the large class of **QUBO** problems).
>
> We will add these insights in the final version.

---

> > ### Author Rebuttal · Reviewer_4ZdA · 2026-04-02
> >
> > I thank the authors for their comment. I would like to point out that local optima is the right notion of a solution concept within this area, which even for simple objectives is a difficult task as evidenced by my comment about PLS vs CLS hardness.
> >
> > The fact that this paper is algorithmic in nature frames the context around the primary result, that is there is no worst-case convergence guarantees of the first-order optimization algorithm, but rather the landscape equivalence between the two problems. For a particular "hard" instance of the problem, it will also probably be difficult for the first-order algorithm to solve it, leading to no contradiction.
> >
> > With this in mind, I will raise my score in accordance with the strong experimental results over multiple settings of interest.

---

> > > ### Author Response · Authors · 2026-04-03
> > >
> > > Thank you for the thoughtful follow-up and for reconsidering your evaluation. We are glad that our clarifications on the PLS vs CLS perspective and the algorithmic nature of our approach addressed your concerns. We appreciate your recognition of the experimental results and your constructive feedback.

---

### Official Review · Reviewer_iKh6 · 2026-03-13

**Soundness:** 3
**Presentation:** 4
**Significance:** 3
**Originality:** 3
**Overall Recommendation:** 5
**Confidence:** 4

**Summary:**

The paper considers the problem of finding minimal energy of Ising models. The paper proposes a minima-preserving continuous relaxation of the problem where the local minima of the relaxation have a one-to-one correspondence to the local minima of the original problem. This allows for an easy implementation of scalable gradient-based optimization methods such as ADAM optimizers.

**Compliance With Llm Reviewing Policy:**

Affirmed.

**Key Questions For Authors:**

Can the authors comment on the SK instance and the method of https://arxiv.org/abs/1812.10897?

While the paper establishes a correspondence between continuous local minima and discrete local minima, it would be interesting to understand how these relate to global optimality. For example, is it possible to bound the error between local minima and global minima for important instances (such as MAX-CUT or NPP) of this problem?

**Limitations:**

The paper would benefit from a more explicit discussion of limitations.

**Strengths And Weaknesses:**

The paper addresses an important problem in combinatorial optimization and general machine learning. It is well written and technically sounds with appropriate structure and decent literature reviews. Both technical and empirical results are well presented and support the main claims of the paper. The paper provides an interesting formulation with rigorous guarantees, allowing the use of scalable gradient-based mainstream optimization. Ultimately, the proposed approach is a heuristic, not always winning against a variety of competing methods, but nonetheless an interesting one. One weakness in the comparison is in the evaluation for spin glasses, namely, the SK model: the comparison methods should include the state of the art (https://arxiv.org/abs/1812.10897) which is message-passing algorithm that achieves the minima.

---

> ### Author Rebuttal · Authors · 2026-03-27
>
> We are glad that the reviewer finds the paper well-written, technically sound, and supported by strong theoretical and empirical results, and we thank them for the helpful suggestions to further strengthen the evaluation.
>
> ---
>
> ### **C1. Comparison with IAMP (SOTA for SK models)**
> We thank the reviewer for pointing out the SOTA baseline for SK models (more precisely, the Gaussian Orthogonal Ensemble (GOE) of random matrices) : Incremental Approximate Message Passing (IAMP) [arXiv:1812.10897]. We benchmark IAMP on both:
>
> * **GOE SK model:** $J_{ij}=J_{ji}\sim\mathcal{N}(0,1/n),J_{ii}\sim \mathcal{N}(0,2/n)$
> * **Standard SK model:** $J_{ij}=J_{ji}\sim \mathcal{N}(0,1),J_{ii}=0$
>
> We use the best IAMP parameters for each model.
>
> For MiP-CRIP, we fix the parameters (tuned on a small 100-spin model) across all instances and scales:
> $\alpha=1.4996\times10^{-5},\beta=0.001,\lambda=0.0707,\sigma=10^{-3},T=10,K=200$, and ADAM parameters: $\tau=1$ (learning rate), $\beta_1=0.09$ (1st moment), $\beta_2=0.999$ (2nd moment).
>
> We quantize $J_{ij}$ to 5 decimals, yielding $\gamma_0=10^{-5}$ and satisfying admissibility:
> $3\beta\lambda^2<\alpha<\beta\lambda^2+\gamma_0$.
>
> Results for 100 instances per case are summarized in **Table R1**.
>
> ### **Table R1: Comparison of IAMP vs MiP-CRIP on GOE and Standard (Std) SK models**
>
> |Type| Spins|Avg Energy|Best Energy|Avg sync|Best sync|Avg Time (s)|
> |-|-|-|-|-|-|-|
> |||(IAMP/MiP-CRIP)|(IAMP/MiP-CRIP)|(IAMP/MiP-CRIP)|(IAMP/MiP-CRIP)|(IAMP/MiP-CRIP)|
> |GOE|100|-62.00/**-71.39**|-70.72/**-77.46**|0.938/**1.000**|**1.000**/**1.000**|0.037/**0.022**|
> |GOE|200|-127.76/**-144.74**|-138.11/**-155.24**|0.941/**1.000**|0.975/**1.000**|0.043/**0.034**|
> |GOE|500|-322.67/**-363.59**|-338.25/**-375.80**|0.944/**1.000**|0.970/**1.000**|**0.071**/0.091|
> |GOE|1000|-643.85/**-728.27**|-664.81/**-750.32**|0.943/**1.000**|0.958/**1.000**|0.179/**0.137**|
> |Std|100|-414.84/**-504.95**|-515.43/**-545.49**|0.915/**1.000**|0.990/**1.000**|0.035/**0.022**|
> |Std|200|-1122.42/**-1438.63**|-1347.49/**-1542.79**|0.900/**1.000**|0.965/**1.000**|0.039/**0.035**|
> |Std|500|-4129.83/**-5752.52**|-4563.74/**-5998.39**|0.880/**1.000**|0.906/**1.000**|**0.060**/0.088|
> |Std|1000|-10895.32/**-16269.86**|-11660.55/**-16767.25**|0.870/**1.000**|0.897/**1.000**|0.144/**0.138**|
>
> **Observation:** MiP-CRIP consistently achieves lower Ising energy than IAMP on both GOE and standard SK models, while converging to valid one-flip local minima (*sync=1*) across all runs.
>
> ---
>
> ### **C2. While the paper ... is it possible to bound the error between local minima and global minima for important instances (such as MAX-CUT or NPP) of this problem?**
> We agree that our guarantees are in terms of **one-flip local optimality**, not global optimality.
>
> While IAMP provides guarantees for a specific SK model (GOE), our method targets the broader class of **general Ising problems**. In this setting, bounding the optimality gap between local and global minima is highly problem-dependent and remains challenging.
>
> Our contribution is complementary:
>
> * We provide a **structure-preserving relaxation** where all continuous local minima correspond exactly to discrete one-flip local minima
> * This enables efficient exploration of a **reduced, structured subset** of the search space that still contains the global optimum
>
> Empirically, MiP-CRIP consistently finds **lower-energy solutions** across diverse problem classes (SK, MAX-CUT, NPP), outperforming both specialized and general-purpose baselines (**Table R1** and *Table 1* from paper). In *Figure 4* for the MAX-CUT problem, MiP-CRIP outperforms GW-SDP, which has an approximation ratio of 0.878, and efficiently matches the ILP-based exact solvers (Gurobi, CP-SAT), which provide a solution certificate.
>
> ---
>
> ### **Summary:**
>
> * Our method beats IAMP (SOTA for SK model) by consistently achieving lower energy while converging to valid local minima (*sync=1*) each time.
> * Our method guarantees **local-minima preservation**, not global optimality, but empirically finds high-quality solutions.
>
> We will be happy to include these in the final version.

---

> > ### Author Rebuttal · Reviewer_iKh6 · 2026-04-06
> >
> > Thank you for the detailed reply. Comparisons in SK models will be the great addition. I have already selected 5 as a score and hence maintaining it.

---

> > > ### Author Response · Authors · 2026-04-06
> > >
> > > Thank you for the encouraging feedback and for maintaining your score. We’re glad the added SK comparison was helpful, and we agree it’s a valuable addition. We will include these results in the final version.

---

### Official Review · Reviewer_HXZ2 · 2026-03-15

**Soundness:** 3
**Presentation:** 3
**Significance:** 2
**Originality:** 2
**Overall Recommendation:** 3
**Confidence:** 4

**Summary:**

In this submission, the authors consider discrete Ising optimization and propose a continuous relaxation framework, namely MiP-CRIP, based on polynomial attractors. The main goal is to address the misalignment between the local minima of smooth relaxations and the original discrete problem. The key novelty is an equivalence theorem proving that, under *admissibility* conditions on the attractor. The authors then translate the theoretical analysis into a practical, scalable, differentiable solver. The method is evaluated on standard benchmarks, including spin-glass, MAX-CUT, and number partitioning. The results demonstrate improved solution quality and better synchronization with valid discrete states compared to existing baselines.

**Compliance With Llm Reviewing Policy:**

Affirmed.

**Final Justification:**

I thank the authors for the detailed and thoughtful rebuttal. I acknowledge that other reviewers may view local optimality as an appropriate solution concept in this area; however, my assessment places more emphasis on stronger guarantees, which I still find somewhat limited. I am not yet convinced that its significance and impact are strong enough for acceptance in its current form. I will maintain my original score.

About the Flag of this paper for an ethics review error: **I sincerely apologize for this error. I corrected it already.**

**Key Questions For Authors:**

Please check the questions in my weak points.

**Limitations:**

yes

**Strengths And Weaknesses:**

Overall, I think the paper considers an important optimization problem. The theoretical contribution of this paper is solid. However, I also have some concerns. The three strong points and weak points are listed as follows:

strong points:

- 1. The theoretical result is interesting and fairly clean. The authors formalize admissibility conditions for the attractor and prove that local minima of the relaxed objective coincide with scaled one-flip local minima of the original Ising problem.

- 2. The proposed method is quite general, which can be applied to different versions of Ising problems. For example, the algorithm can be applied to MAX-CUT and NPP, rather than being tailored to only one application.

- 3. The experimental results are competitive compared with baseline methods.  The efficiency of the proposed method is promising as shown in Figure 4.

Weak points:

- 1. To me, the main contribution is mainly in the local minima sense. The theory preserves one-flip local optimality, which is useful, but this is still a relatively weak notion compared with approximation global guarantees. If the local minima are bad, then the theoretical guarantee also could be very bad.

- 2. There are several parameters in the proposed algorithm ($\alpha, \beta, \lambda$). The authors may need to justify how hard to specify these parameters in a reasonable and efficient way.

- 3. The improvements of the proposed method are not very significant. The authors show the results in Table 1. However, it seems that the FEM method is the most efficient. Also, these datasets are not large-scale. I was wondering the efficiecny can be achieved in large-scale problems.

---

> ### Author Rebuttal · Authors · 2026-03-27
>
> We thank the reviewer for the careful reading, for acknowledging the importance of the problem and the clarity of our theoretical contributions, and for the constructive feedback on optimality, efficiency and scalability. We address these points below.
>
> ---
>
> ### **C1. To me, the main contribution is mainly in the local minima sense. The theory preserves one-flip local optimality, which is useful, but this is still a relatively weak notion compared with approximation global guarantees. If the local minima are bad, then the theoretical guarantee also could be very bad.**
>
> We agree that one-flip local optimality is weaker than global approximation guarantees, and we do not claim otherwise. Finding global guarantees or approximation ratios for general Ising problems is combinatorially hard and is beyond the scope of this work.
>
> Our focus is on reliably identifying the **discrete local minima**. This enables efficient exploration of a significantly reduced subset of the large search space that still contains the global optimum. The formulation naturally enables the use of efficient first-order optimizers from smooth optimization, such as ADAM, for tackling combinatorial optimization problems.
>
> We further note that the admissibility conditions in *(28)* define a **parameter region** within which this equivalence holds. In practice, searching (parameter tuning) within this region allows us to identify **high-quality (low-energy) local minima**.
>
> Empirically, we observe that these local minima are competitive and often outperform existing SOTA methods across multiple Ising problem classes (**Table R1**, *Figure 3* and *Table 1* ). In *Figure 4*, it matches ILP-based exact solvers (Gurobi, CP-SAT) quite efficiently, attaining near-global solutions.
>
>
> ---
>
> ### **C2. There are several parameters in the proposed algorithm ($\alpha, \beta, \gamma$). The authors may need to justify how hard to specify these parameters in a reasonable and efficient way.**
>
> Continuing from the previous response, we agree that parameter selection is important for performance and to find high-quality local minima. In our case, parameter tuning is not a bottleneck, as we have used an adaptive grid search method (described in *Appendix E* of the paper) with *3 to 5 values per parameter for efficiency*.
>
> Also, we need to tune the parameters only once for a particular class of Ising models (e.g. SK model, K2000), then the same set of parameters gives reasonably better solutions than other solvers for different instances within that class (as seen from *Figure 3* and *Table 1* from the paper).
>
> Another practical strategy is to tune parameters on a small-scale instance of the same class (which can be done in real time) and reuse them for larger-scale problems. The effectiveness of this transfer is demonstrated in **Table R1** (response to reviewer **iKh6**).
>
> ---
>
> ### **C3. The improvements of the proposed method are not very significant. The authors show the results in Table 1. However, it seems that the FEM method is the most efficient. Also, these datasets are not large-scale. I was wondering the efficiecny can be achieved in large-scale problems.**
>
> We thank the reviewer for highlighting this point. While FEM performs well on certain SK instances, it does not generalize as effectively across broader Ising problem classes (espacially NPP). Our method is designed as a **general-purpose Ising solver**, and demonstrates consistently strong performance across diverse settings. Also, the runtimes in *Table 1* don't include parameter tuning time for FEM, which is more than ours due to a larger number of hyperparameters.
>
> Regarding scalability:
>
> * Our method is a **first-order optimization algorithm** with updates dominated by matrix-vector multiplications, making it **GPU-friendly and scalable**.
> * The datasets used (e.g., G-set, SK, K2000) are standard benchmarks in the Ising/MAX-CUT literature, enabling fair comparison with prior work.
> * For larger-scale problems, parameter tuning seems the main concern, but as stated in our comments to **C2**, this can be amortized across scales.
>
> ---
>
> ### **Summary:**
>
> * The method guarantees **local-minima preservation**, not global approximation, while consistently achieving lower-energy solutions and outperforms strong baselines across problem classes.
> *  Hyperparameter tuning is efficient, amortizable across problem classes and scales, and compatible with automated methods.
> * The method is **scalable and general-purpose**, with strong empirical performance across diverse Ising problems.
>
> We will clarify these points and emphasize scalability in the final version.
>
> ---
>
> **Lastly, we want to ask the reviewer to kindly clarify the reason for selecting “Flag this paper for an ethics review”, given that no specific ethical concerns were indicated.**

---

> > ### Author Rebuttal · Reviewer_HXZ2 · 2026-04-04
> >
> > I thank the authors for the detailed and thoughtful rebuttal. I acknowledge that other reviewers may view local optimality as an appropriate solution concept in this area; however, my assessment places more emphasis on stronger guarantees, which I still find somewhat limited. I am not yet convinced that its significance and impact are strong enough for acceptance in its current form. I will maintain my original score.
> >
> > About the Flag of this paper for an ethics review error: **I sincerely apologize for this error. I will correct it as soon as possible.**

---

> > > ### Author Response · Authors · 2026-04-04
> > >
> > > We thank the reviewer for the thoughtful follow-up and for acknowledging our detailed rebuttal. We hope our earlier responses have clarified the questions regarding efficient hyperparameter tuning and performance across general Ising problems, and we have also provided empirical evidence on the effectiveness of the resulting one-flip local minima.
> > >
> > > Here, we would like to clarify the theoretical scope and significance of our work:
> > >
> > > * **Stronger guarantees:**
> > > For general Ising/QUBO problems, stronger guarantees, such as approximation ratios or $(1-\epsilon)$-optimality, are not known. Existing results apply only to specific structured cases (e.g., GW-SDP for MAX-CUT with non-negative weights, or IAMP for SK models with GOE structure). In this sense, our guarantee is **one of the few general guarantees applicable across arbitrary Ising models**.
> > >
> > > * **Novelty of our result:**
> > > Our main contribution is a **landscape equivalence theorem**, which establishes an exact correspondence between continuous local minima and discrete one-flip local minima. This is novel in the sense that it eliminates spurious minima in continuous relaxations while preserving the combinatorial structure of the problem, something not achieved by standard relaxations. As a result, it enables the use of scalable first-order continuous optimization methods to reliably recover valid discrete solutions.
> > >
> > > * **The significance of one-flip local minima:**
> > > The notion of one-flip local minima corresponds to **valid combinatorial structures** across problem classes e.g., feasible configurations in MAX-CUT, NPP (also for other graph-theoritic problems like the Maximum Independent Set Problem, it corresponds to independent sets, for the Maximum Clique Problem, it corresponds to a valid click, for the Traveling Salesman Problem, it corresponds to a valid Hamiltonian path). Our result ensures that such structures can be obtained via **continuous optimization**, avoiding the scalability limitations of traditional combinatorial methods (e.g., ILP, dynamic programming, branch-and-bound).
> > >
> > > * **Stronger local minima notions:**
> > > Extending guarantees to $k$-flip local minima becomes increasingly restrictive: even modest extensions (e.g., $k=2$) significantly constrain the class of admissible Ising models, while guarantees approaching global optimality (i.e. $n$-flip local minima) are unlikely in general (unless major complexity assumptions are resolved). This highlights that **one-flip local optimality is a natural and broadly applicable level of guarantee**.
> > >
> > > We will carefully incorporate these clarifications in the revision to more clearly emphasize the theoretical contribution and its scope.

---

### Official Review · Reviewer_UYjV · 2026-03-22

**Soundness:** 2
**Presentation:** 2
**Significance:** 3
**Originality:** 3
**Overall Recommendation:** 4
**Confidence:** 1

**Summary:**

The paper studies the problem of solving general Ising (QUBO) optimization via a continuous relaxation that preserves the discrete structure of the problem at the level of local optimality. The main result shows that, under suitable conditions on the choice of an attractor function and parameters, there is a one-to-one correspondence between the local minima of the continuous objective and the one-flip local minima of the original discrete problem. In particular, the authors design a polynomial relaxation (MiP-CRIP) whose minimas align exactly with discrete solutions, thereby avoiding spurious local minima that typically arise in standard relaxations. Building on this, they propose a gradient-based optimization method that provably converges to discrete local minima while operating in the continuous domain. Empirically, the approach achieves competitive performance on standard benchmarks such as MAX-CUT and spin glass instances, often matching or outperforming existing heuristics while offering improved scalability.

My main concern is that the theoretical guarantees rely on a robustness condition involving a global margin parameter \gamma (see definition  (19)), which captures the minimum discrete improvement under single-bit flips. While the authors provide lower bounds on \gamma in finite-precision settings, computing or tightly estimating this quantity is itself intractable in general, and it appears to me that the correctness of the reduction hinges on choosing parameters that depend on it.  As a result, the framework appears to shift the core difficulty of the problem into verifying or implicitly satisfying this condition, which is handled heuristically via parameter tuning in practice. Additionally, the focus on one-flip local minima only provides local optimality and it is unclear whether one can use it to find global optimal solutions.

**Compliance With Llm Reviewing Policy:**

Affirmed.

**Final Justification:**

The authors responses were satisfactory. I still think the primary contribution of this paper is on the practical side and the authors back it up with strong experiments on benchmarks. Consequently, I've raised my score. I maintain a low confidence due to a lack of experience on the practical side.

**Key Questions For Authors:**

NA

**Limitations:**

yes

**Strengths And Weaknesses:**

From a theoretical perspective, the reliance on \gamma makes the result less compelling to me. I should note, however, that I do not have significant experience on the practical aspects of working at the interface of discrete and continuous optimization, so I may be underestimating the practical relevance of this contribution. Accordingly, I assign my rating with low confidence.

---

> ### Author Rebuttal · Authors · 2026-03-27
>
> We thank the reviewer for the detailed summary of our work and for recognizing its practical relevance and scalability, and we appreciate the thoughtful concerns regarding the theoretical aspects. We address these concerns below.
>
> ---
>
> ### **C1. My main concern is that the theoretical guarantees rely on a robustness condition involving a global margin parameter \gamma (see definition (19)), ... As a result, the framework appears to shift the core difficulty of the problem into verifying or implicitly satisfying this condition, which is handled heuristically via parameter tuning in practice.**
>
> We agree that computing the exact margin $\gamma$ from *(19)* is combinatorial. But, our theory **does not require the exact value of $\gamma$**. To satisfy the admissibility conditions in *(28)*, only a valid lower bound ($\gamma_0 \le \gamma$) is sufficient.
>
> Such lower bounds are **readily available and inexpensive** to obtain.
>
> * As shown in *Proposition 3.4*, one immediate choice is the trivial bound given by the least significant bit ($\gamma_0 = lsb(J)$)
> * In practical settings discussed in *Remark 3.5*, most of the graph-based combinatorial problems involve integers or low-precision weights, so a valid lower bound is either directly available or can be chosen conservatively.
> * Even in more extreme cases, one can set the bound adaptively so that admissibility is eventually guaranteed. (e.g. $\gamma(t) = \gamma_0^t \to 0$).
>
> Therefore, the framework does **not move the computational burden to estimating the exact margin $\gamma$**. Instead, it relies on **simple, conservative bounds** that suffice for all theoretical guarantees.
>
> This is validated in our experiments. For example, in **Table R1** (response to reviewer **iKh6**), with the stated precision and lower-bound choice, the admissibility condition is consistently satisfied, and the method converges to one-flip local minima (*sync* = 1) across runs.
>
> ---
>
> ### **C2. Additionally, the focus on one-flip local minima only provides local optimality and it is unclear whether one can use it to find global optimal solutions.**
>
> We agree that one-flip local minima only provide **local optimality**, and we do not claim global optimality guarantees (finding the global optimal solution is NP-hard in general).
>
> The global optimum lies within the set of local minima, which is much smaller than the full exponential search space. Our relaxation preserves this structure (*Theorem 3.13*), enabling efficient first-order methods (e.g., ADAM) to explore the discrete local-minima landscape.
>
> Empirically (in *Figure 4*), it finds the near-global solutions achieved by ILP solvers (Gurobi, CP-SAT) quite efficiently.
>
> Efficient hyperparameter tuning ensures these good-quality solutions. For this, we request the reviewer to check our comments to **C1, C2** of reviewer **HXZ2**, where we have addressed this in detail.
>
> ---
>
> In summary:
>
> * The method does *not* require computing the exact margin $\gamma$; simple lower bounds (e.g., $lsb$) suffice and are trivial to obtain in practice.
> * The method targets **discrete one-flip local minima**, not global optima.
> * The method efficiently explores a reduced, structured space of discrete local minima using scalable first-order continuous optimizers.
> * The quality of local minima is ensured by efficient hyperparameter tuning.
>
> We will clarify these points in the final version.

---

> > ### Author Rebuttal · Reviewer_UYjV · 2026-04-03
> >
> > Thank you for providing a clarification. I am happy with the responses and will raise my score.

---

> > > ### Author Response · Authors · 2026-04-04
> > >
> > > Thank you for taking another look and for your kind update. We’re glad the clarifications helped address your concerns, and we appreciate your revised evaluation.

---

### Decision · Program_Chairs · 2026-04-30

**Decision:**

Accept (regular)

**Comment:**

The paper considers the Ising optimization problem and proposes a novel approach, namely finding local minima via a fractional relaxation of the problem. The main result establishes a correspondence between the local minima of the fractional relaxation and the 1-neighborhood local minima of the discrete problem. Based on this result, the authors propose a gradient-descent-type algorithm for solving such Ising minimization problems.

All reviewers (and I) agree that the approach is novel, the idea is interesting, and the correspondence result is indeed surprising. However, some reviewers expressed concerns regarding both the significance of the results (i.e., the focus on local minima) and the extent of the experimental evaluation, including the difficulty of the considered datasets. Despite the authors’ responses, some of these concerns remain not fully addressed, in my opinion.

Additionally, although not explicitly raised by the reviewers, I note that the same guarantee—convergence to a local neighborhood minimum—can be achieved by a trivial local search algorithm in the discrete space. In this sense, it remains unclear what additional benefits the proposed approach provides, given that the same guarantee can be obtained via simpler methods.
Overall, I believe the paper has clear merits, but another round of revision would be beneficial before publication.